# Liquid biopsy for the diagnosis of EBV-positive Burkitt's lymphoma in endemic areas

Burkitt's lymphoma (BL) is common in sub-Saharan Africa, yet diagnosis is often delayed due to limited pathology capacity. Here we evaluated blood-based liquid biopsies from 377 children and young adults with clinically suspected lymphoma at four hospitals in Tanzania and Uganda, assessing diagnostic accuracy and turnaround time (TAT). After extensive pathology capacity building, a gold-standard diagnosis was established using tissue morphology, a limited validated immunohistochemistry panel and independent dual histopathologist review. Using clinical features and circulating tumor DNA markers (*MYC* mutations, *MYC*–immunoglobulin translocations and Epstein–Barr virus fragmentomics), we trained six penalized logistic regression models with tenfold crossvalidation ($n = 212$). The best-performing model was externally validated in a prospective real-world cohort ($n = 56$). Diagnostic accuracy, yield and TAT were compared head to head between liquid biopsy and the gold standard in 58 participants. The comprehensive model achieved the highest performance (area under the curve (AUC) 0.95, 95% confidence interval (95% CI) 0.901–0.981, sensitivity 0.86, specificity 0.95), confirmed by external validation (AUC 0.98, 95% CI 0.942–1.000). Liquid biopsy was the only diagnostic result available at the multidisciplinary review in 42% of participants and reduced median diagnostic TAT from 46.8 d to 6.5 d ($P = 4.42 \times 10^{-10}$). These findings demonstrate that liquid biopsy enables fast, highly accurate molecular diagnosis of EBV$^+$ BL and may substantially reduce treatment delays in resource-limited settings.

Burkitt's lymphoma (BL) is a common cancer in children, with an aggressive course, but a high cure rate when treatment is initiated early and correctly[1,2]. It is associated with Epstein–Barr virus (EBV) infection in 95% of cases in sub-Saharan Africa (SSA) and shows a strong correlation with malaria endemic areas[3–5]. To differentiate BL from other non-Hodgkin's lymphomas, standard-of-care (SoC) diagnosis requires the integration of morphology, immunophenotype and demonstration of a *MYC* rearrangement in the absence of BCL2 and/or BCL6 rearrangements[6]. However, these methods are not readily available in many low-income countries, where the burden of BL is highest and the availability of timely, reliable pathology services remains an important

challenge[7–9]. In addition to a shortage of histopathologists, access to immunohistochemistry (IHC) is limited and, even when present, often inconsistently performed due to difficulties maintaining a stable supply of reagents and antibodies[9–12]. As a result, diagnosis typically relies solely on hematoxylin and eosin (H&E) morphology, making precise differential diagnosis impossible. Moreover, delays in diagnosis and treatment initiation remain a critical barrier to care, with a median delay of 91 d for BL from first onset of symptoms and a 51% probability of treatment not starting until 90 d after presentation[9].

A three-phase diagnostic scoring system, based on a limited IHC panel, has been proposed to support diagnosis in environments where

✉e-mail: clarachamba@gmail.com

IHC is either unavailable or inconsistently implemented. This scoring system relies on a minimal set of markers that are more accessible in low-resource laboratories[12]. This scoring system, designed for settings with limited pathology infrastructure, demonstrated 81% concordance with SoC BL pathology diagnosis, supporting its use as a practical and reliable reference standard in such contexts.

New methodologies are continually emerging in the field of liquid biopsies to better define and expand their clinical utility[13–16]. In principle, targeted sequencing of circulating cell-free DNA (cfDNA) extracted from plasma enables identification of genetic aberrations associated with different tumor types[17,18]. However, for SSA, the application of cfDNA analysis in cancer care is an under-researched topic. Despite a plethora of literature from high-income countries on the utility of this approach for the detection of minimal residual disease and accelerated approvals by the US Food and Drug Administration of some early cancer detection tests[13,19], there is still a paucity of data on its potential applications in SSA.

We previously proposed a liquid biopsy diagnostic model as a minimally invasive yet accurate modality for the detection of BL in children and young adults with suspected lymphoma[20]. In this preliminary analysis involving 20 children, key features, including the hallmark *MYC*–immunoglobulin (*MYC*–Ig) translocation, acquired *MYC* intron 1 single nucleotide variants (SNVs), tumor fraction of circulating cfDNA (ctDNA) and autosomal DNA entropy, were significantly associated with BL diagnosis[20]. When integrated into a penalized logistic regression model, these variables demonstrated strong discriminative performance, achieving an area under the curve (AUC) of 0.95 for the detection of BL[20]. However, the limited targeted panel used detected the *MYC*–Ig translocation in only 50% of cases of BL, highlighting the need for additional molecular markers to further improve diagnostic accuracy.

Genetic alterations in BL include *ID3* mutations, present in up to 76% of cases[21,22], which disrupt cell cycle regulation, and *TP53* mutations, found in up to 58% of cases[22–24], which impair key tumor-suppressor functions. Together SNVs and insertions and/or deletions in *MYC*, *ID3* and *TP53* are among the most frequently reported mutated genes in BL and are recognized as key contributors to BL lymphomagenesis[22,25]. Assessing the median variant allele frequency (VAF) of mutations in *MYC*, *ID3* and *TP53* could offer complementary molecular insights and improve the identification of cases of BL beyond reliance on *MYC* translocation status. In addition, given that >95% of cases of BL are EBV⁺, it is important to consider whether EBV DNA characteristics, such as fragment size and relative abundance, could meaningfully enhance the performance of diagnostic models. For example, in nasopharyngeal carcinoma, a blood-based screening test combining EBV DNA quantity expressed as the EBV proportion (EBVP) with EBV fragment size ratios, achieved superior diagnostic performance compared to using EBV quantity alone[26], underscoring the value of integrating multiple molecular parameters to improve the accuracy and predictive power of diagnostic assays.

Here we present the diagnostic accuracy study of this liquid biopsy approach in a larger cohort of children and young adults with suspected lymphoma incorporating multiple molecular attributes, followed by an independent, prospective, real-world evaluation, to provide further evidence of its clinical utility. Overall diagnostic performance was assessed comprehensively based on accuracy (sensitivity, specificity), diagnostic yield and TAT, compared to the best local gold-standard pathology (GSP) as previously described[12].

## Results

### Phase I: clinical validation of liquid biopsy compared to GSP tissue biopsy as the gold standard

In phase I of the study, we enrolled a total of 313 children and young adults clinically suspected of having a diagnosis of lymphoma (Extended Data Fig. 1). A tissue biopsy was collected and processed for a histopathology diagnosis with H&E staining in 89.5% (280 out of 313) of these patients. For the remaining 33 participants, 8 died before undergoing a tissue biopsy, 5 declined the procedure or were lost to follow-up and 20 had inadequate biopsy samples for histopathological assessment. Among the 280 samples, only 212 (67.7% of the total; 212 out of 313) achieved a GSP tissue biopsy diagnosis consisting of IHC with a limited panel[12] and review by at least 2 pathologists. The remaining 68 samples were not of adequate quality or sufficient quantity to allow IHC. The failure rate of H&E diagnosis was 10.5% (33 out of 313), whereas the failure rate of GSP tissue biopsy was 24.3% (68 out of 280).

Liquid biopsy was collected from all the 313 participants. However, sequencing was prioritized for samples with a GSP tissue biopsy because this was the gold-standard comparator in the clinical validation. In total, samples from 247 participants were sequenced. Although a total of 4 runs (covering 24 samples) initially failed during the first attempt, all were successfully repeated and sequencing results were subsequently generated for all samples. For clinical validation of the liquid biopsy test, we analyzed data from the 212 samples in this cohort with a GSP tissue biopsy diagnosis (Extended Data Fig. 1).

### Clinical characteristics

For the phase I enrollment, participants had a median age of 13 years (interquartile range (IQR) 9–17 years), with a male-to-female ratio of 2:1. Eleven (3.5%) participants had a serologically confirmed HIV diagnosis and were on antiretroviral treatment, whereas nine (2.9%) had a history of having been diagnosed and treated for tuberculosis. Only 15 (4.8%) participants reported a positive family history of cancer in their family. A jaw mass was the presenting feature in 25% (78 out of 313) of participants, peripheral lymph node enlargement was present in 68% (213 out of 313) and at least half (53%, 167 out of 313) the participants had an abdominal organomegaly. Participants had a median hemoglobin of 9.81 g dl⁻¹ (IQR 8.15–11.30 g dl⁻¹) and median lactate dehydrogenase (LDH) levels of 679 IU l⁻¹ (IQR 393–1,233 IU l⁻¹). The most common GSP tissue biopsy diagnosis was classic Hodgkin's lymphoma (HL) (26%, 80 out of 313), followed by BL (25%, 77 out of 313) and diffuse large B cell lymphoma (DLBCL) (14%, 45 out of 313). Other childhood cancers were present in 9.6% (30 out of 313), whereas 9.3% (29 out of 313) of participants had benign conditions. The demographic, clinical and laboratory characteristics of participants for both phases of the study are summarized in Extended Data Tables 1 and 2.

### Associations of demographic, clinical and liquid biopsy variables and a diagnosis of BL

Among the 212 participants, where GSP tissue biopsy diagnosis was achieved, 38.2% (81 out of 212) had a diagnosis of BL and 61.8% (131 out of 212) had a non-BL diagnosis. For the non-BL participants, 36.7% (48 out of 131) had HL, 16.8% (22 out of 131) had DLBCL, 18.3% (24 out of 131) had a benign diagnosis (benign tumor, tuberculous adenitis, reactive or EBV⁺ lymphadenitis) and the remaining 28.2% (37 out of 131) had a mixture of other types of cancer. A younger age, presence of an abdominal or jaw mass and elevated LDH were significantly associated with a diagnosis of BL ($P < 0.05$ for all; Table 1). For the liquid biopsy variables, diagnosis of BL was associated with higher ctDNA levels, greater median number of mutations in *MYC* intron 1, increased *EBER1*, *EBER2* and *EBNA2* copies per cell, higher EBV DNA fragment size ratio and EBVP, increased EBV DNA fragment size entropy and autosomal fragment size entropy and the presence of a *MYC*–Ig translocation ($P < 0.01$ for all; Table 1 and Extended Data Fig. 2). Additional clinical and molecular characteristics are reported in Supplementary Table 1.

*MYC*–Ig translocations were detectable in 48% of confirmed cases of BL using our limited targeted sequencing panel. Most *MYC*–*IGH* breakpoints on the *MYC* locus occurred within the class II (46%, 18 out of 39) and class III (28%, 11 out of 39) regions. On the *IGH* locus, 51% (20 out of 39) of breakpoints were located within the VDJ junction region (Extended Data Fig. 3). As mutations in *MYC* have previously been

**Table 1 | Clinical and liquid biopsy characteristics of phase I participants stratified as BL and non-BL (*n*=212 children and young adults with clinically suspected lymphoma)**

| Characteristic | Overall[a] | BL[a] | Non-BL[a] | P value[b] | Test |
|---|---|---|---|---|---|
| Age, years | 10.3 (6.8, 14.1) | 9.9 (6.8, 12.5) | 10.7 (6.9, 15.7) | 0.0429 | Wilcoxon's |
| Tumor site | | | | $5.43 \times 10^{-16}$ | $\chi^2$ |
| Jaw or abdominal, *n* (%) | 103 (49) | 68 (84) | 35 (27) | | |
| Other, *n* (%) | 109 (51) | 13 (16) | 96 (73) | | |
| LDH, IU l$^{-1}$ | 656 (385, 1,312) | 945 (553, 1,633) | 516 (343, 824) | $7.50 \times 10^{-6}$ | Wilcoxon's |
| CtDNA, hGE l$^{-1}$ | 153 (0, 664) | 530 (166, 1,579) | 28 (0, 241) | $9.93 \times 10^{-12}$ | Wilcoxon's |
| *MYC* intron 1 mutation count | 1 (0, 13) | 15 (6, 28) | 0 (0, 1) | $2.03 \times 10^{-19}$ | Wilcoxon's |
| *EBER1* (copies per cell) | 0.1 (0, 3.1) | 3.1 (0.1, 6.6) | 0 (0, 0.3) | $2.00 \times 10^{-12}$ | Wilcoxon's |
| *EBER2* (copies per cell) | 0.07 (0, 2.47) | 3.06 (0.15, 5.51) | 0 (0, 0.22) | $2.10 \times 10^{-12}$ | Wilcoxon's |
| *EBNA2* (copies per cell) | 0 (0, 2.6) | 1.2 (0.0, 10.1) | 0 (0, 0.2) | $2.41 \times 10^{-6}$ | Wilcoxon's |
| EBV$_{max}$ (copies per cell) | 0.1 (0, 5.2) | 5.3 (0.4, 11.3) | 0 (0, 0.4) | $3.07 \times 10^{-12}$ | Wilcoxon's |
| EBV fragment size ratio | 0.43 (0, 0.67) | 0.63 (0.40, 0.78) | 0.15 (0, 0.55) | $7.97 \times 10^{-8}$ | Wilcoxon's |
| EBVP | 0 (0, 0.008) | 0.008 (0, 0.019) | 0 (0, 0.001) | $4.71 \times 10^{-13}$ | Wilcoxon's |
| EBV fragment entropy, bits | 4.64 (1.10, 5.22) | 5.25 (4.88, 5.49) | 2.43 (0, 4.90) | $1.054.71 \times 10^{-13}$ | Wilcoxon's |
| *MYC*–Ig translocation, *n* (%) | 39 (18) | 39 (48) | 0 (0) | $1.474.71 \times 10^{-18}$ | $\chi^2$ |

[a]Median (quartile (Q)1, Q3) for continuous, *n* (%) for categorical. [b]Exact *P* values from two-sided Wilcoxon's rank-sum or $\chi^2$ tests. hGE, haploid genome equivalents.

reported as surrogate markers for *MYC* rearrangement, we also examined their association with BL in our dataset. Although *MYC* mutations were present in both BL and non-BL samples, the distribution differed markedly between the two groups (Extended Data Figs. 4d,e and 5). BL samples showed a significantly higher mutation burden, reflected by a higher median number of mutations in *MYC* intron 1 and exon 2 compared to non-BL samples (Supplementary Table 1). A fourfold difference was also observed in the median VAF between the BL and non-BL samples. The VAF distribution by diagnosis for each sample is shown in Extended Data Fig. 4c.

### Diagnostic performance of the six models

Six models were constructed using different combinations of variables (Table 2): a clinical model comprising clinical parameters only; an EBV quantitative model based on quantitative (q)PCR-derived metrics; an EBV quantitative plus clinical model; an EBV model incorporating additional qualitative EBV features (fragment size ratio and fragment entropy); a liquid biopsy model including cfDNA-derived variables only; and a comprehensive model combining liquid biopsy and clinical variables. Tenfold crossvalidation showed that all models demonstrated good discriminative ability for BL (AUC ≥ 0.8), with the liquid biopsy model and the comprehensive model showing excellent discriminative ability[27] with AUC values of 0.92 and 0.94, respectively. The sensitivity of the models ranged from 0.57 (EBV quantitative model) to 0.86 (comprehensive model), whereas specificity ranged from 0.78 (clinical model) to 0.95 (liquid biopsy and comprehensive model). Quantification of EBV in copies per cell, which was previously shown to correlate strongly with EBV DNA qPCR[20], was performed inferiorly to the liquid biopsy model both when clinical parameters were included (AUC 0.91 versus 0.95) and when they were not (AUC 0.80 versus 0.92). Overall, the comprehensive model demonstrated the best performance, with an AUC of 0.95, sensitivity of 0.86 and specificity of 0.95 (Fig. 1a,b). Among 81 patients with a confirmed local gold-standard diagnosis of BL, 11 (13.6%) were not detected by the liquid biopsy model (false negatives), whereas 70 (86.4%) were correctly identified (true positives). False-negative cases were comparable to true positives with respect to age, sex distribution and LDH levels. However, they were significantly less likely to present with jaw or abdominal tumors (*P* < 0.05) (Supplementary Table 2).

**Table 2 | BL diagnostic models**

| Model name | Variables |
|---|---|
| Clinical model | Age, sex, duration of presenting illness, site of tumor and LDH |
| EBV quantitative model | *EBER1*, *EBER2*, *EBNA2*, EBV$_{max}$ (maximum EBV copies per cell) |
| EBV quantitative + clinical model | Age, sex, duration of presenting illness, site of tumor, LDH, *EBER1*, *EBER2*, *EBNA2*, EBV$_{max}$ |
| EBV model | *EBER1*, *EBER2*, *EBNA2*, EBV$_{max}$, EBV size ratio, EBVP, EBV entropy |
| Liquid biopsy model | CtDNA, median VAF, *MYC* intron 1 mutations, *MYC* exon 2 mutations, *EBER1*, *EBER2*, *EBNA2*, EBV$_{max}$, EBV size ratio, EBP, EBV entropy, autosomal entropy and *MYC*–Ig translocation |
| Comprehensive model | Age, sex, duration of presenting illness, site of tumor, LDH, ctDNA, median VAF, *MYC* intron 1 mutations, *MYC* exon 2 mutations, *EBER1*, *EBER2*, *EBNA2*, EBV$_{max}$, EBV size ratio, EBVP, EBV entropy, autosomal entropy and *MYC*–Ig translocation |

The relative contribution of each predictor to the comprehensive model is shown in Fig. 1c. Among the included variables, *MYC*–Ig translocation, EBVP, tumor site and autosomal entropy emerged as the strongest predictors, reflected by their large absolute coefficients in the least absolute shrinkage and selection operator (LASSO) regression analysis. Most important predictors for the other models are included in Extended Data Fig. 6.

### Phase II: real-world evaluation of liquid biopsy

In the phase II arm, we enrolled 64 participants, of whom 1 patient died and another was lost to follow-up before the collection of a tissue or liquid biopsy, leaving 62 patients eligible for analysis. With respect to tissue biopsy, GSP tissue biopsy diagnosis was not obtained for six patients: one patient sample was inadequate for processing and five patient samples were insufficient for IHC. In addition, the liquid biopsy sample for one patient was not sequenced because it hemolyzed, disqualifying it for downstream analysis. Consequently, a GSP tissue biopsy

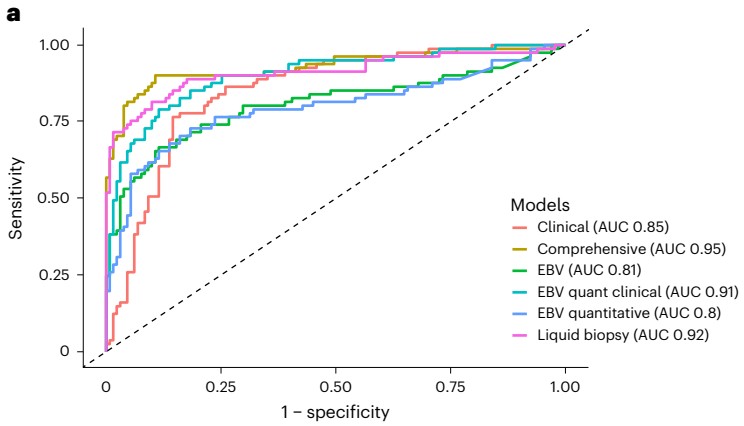

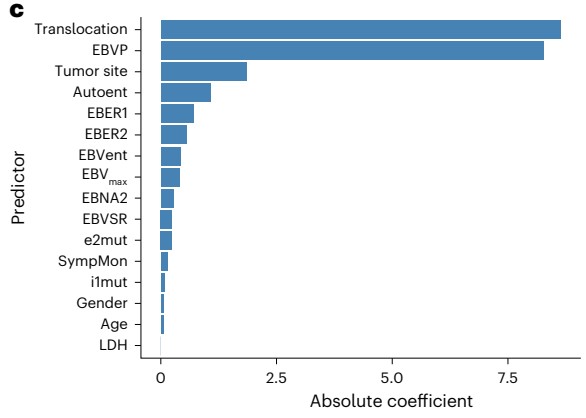

**b**

| Model | Sensitivity (95% CI) | Specificity (95% CI) | AUC (95% CI) | P value |
|---|---|---|---|---|
| Clinical | 0.815 (0.713–0.892) | 0.779 (0.698– 0.846) | 0.851 (0.799–0.904) | – |
| Comprehensive | 0.864 (0.770–0.930) | 0.954 (0.903–0.983) | 0.945 (0.909–0.981) | $4.05 \times 10^{-5}$ |
| EBV quant clinical | 0.778 (0.672–0.863) | 0.878 (0.809–0.929) | 0.908 (0.866–0.951) | $1.63 \times 10^{-3}$ |
| EBV quantitative | 0.568 (0.453–0.678) | 0.947 (0.893–0.978) | 0.796 (0.727–0.865) | $1.66 \times 10^{-1}$ |
| EBV | 0.593 (0.478–0.701) | 0.916 (0.855–0.957) | 0.808 (0.743–0.878) | $3.17 \times 10^{-1}$ |
| Liquid biopsy | 0.741 (0.631–0.832) | 0.969 (0.924–0.992) | 0.917 (0.875–0.959) | $2.32 \times 10^{-2}$ |

**Fig. 1 | Performance and feature importance of diagnostic models for BL.**
**a**, ROC curves comparing clinical, EBV-based, liquid biopsy and comprehensive models. AUCs are shown in the figure. **b**, Sensitivity, specificity and AUC (95% CIs) for each model. *P* values were derived from two-sided DeLong's tests comparing each model against the clinical model; no adjustment for multiple comparisons was applied. **c**, Most influential predictors from the LASSO comprehensive model ranked by absolute coefficient magnitude; coefficients reflect standardized variables (*n* = 212). autoent, autosomal fragment entropy; e2mut, *MYC* exon 2 mutations; EBVent, EBV fragment entropy; i1mut, *MYC* intron 1 mutations; SympMon, duration of presenting symptoms in months.

diagnosis was reached for 56 patient tissue biopsy samples, whereas a diagnostic report was generated for 61 patient liquid biopsy samples (Extended Data Fig. 1). We conducted an external validation of our best-performing model (comprehensive model) on these 56 patients from the phase II cohort after addressing missing values by appropriate imputation methods. The comprehensive model demonstrated excellent discriminative ability, with an AUC of 0.97 (95% CI 0.924–1.000), a sensitivity of 0.94 (95% CI 0.698–0.998) and a specificity of 0.85 (95% CI 0.702–0.943) (Fig. 2a). When the analysis was focused on the 44 patients with a complete dataset, the AUC was 0.97 with a sensitivity of 0.93 (95% CI 0.661–0.998) and a specificity of 0.90 (95% CI 0.735–0.979) (Fig. 2b).

**Integration of the liquid biopsy test into the diagnostic pathway through an MDT meeting**
Diagnosis of the phase II samples was made following the multidisciplinary team (MDT) decision tree (Fig. 3a). Liquid biopsy was the only test available for making a diagnosis in 42.6% (26 out of 61) of cases (Fig. 3b) at the first MDT meeting. Among all the cases that were finally diagnosed as BL (*n* = 15), in the first MDT meeting, 1 (6.7%) was diagnosed using tissue biopsy alone, 6 (40%) were diagnosed with both liquid and tissue biopsy available, whereas 8 (53.3%) were diagnosed using liquid biopsy alone. One child with BL was not diagnosed in the first MDT meeting, but was confirmed in subsequent MDT meetings (Fig. 3b). Six samples were diagnosed with tissue biopsy alone at the first MDT meeting, out of which five were non-BL. This demonstrates that liquid biopsy enabled diagnosis in an additional 53.3% (8 out of 15) of cases of BL at the first MDT meeting, increasing the overall diagnostic yield of BL to 93.3% (14 out of 15), underscoring its potential as a complementary and timely diagnostic tool, especially where tissue biopsy access is limited or delayed. The percentage of occasional decisions taken based on each pathway are denoted on Fig. 3a. In 16.4% (10 out of 61) of cases, neither the liquid biopsy nor the tissue biopsy was

available at the first MDT meeting and diagnosis was therefore made in the subsequent MDT meeting (Fig. 3a).

**TAT for the liquid biopsy and tissue biopsy tests**
TAT was assessed in a head-to-head comparison for 58 samples from children and young adults with suspected lymphoma who had complete data for both the liquid and the tissue biopsies. The median time from patient presentation at the cancer center to tissue biopsy sample collection was 7.3 (IQR 2.6–13.6 d). The median time from sample collection to its receipt at the histopathology lab was 1 d (IQR 0.5–2.2 d), whereas sample processing to the generation of an H&E slide ready for interpretation took 3.5 d (IQR 1.5–5.6 d). The median time for H&E slide reporting was 3.7 d (IQR 1.7–8.6 d), whereas tissue processing for GSP tissue biopsy diagnosis had a median TAT of 43 d (IQR 6.6–207.4 d) (Fig. 4a).

TAT for liquid biopsy processing was assessed from the time that the sample arrived at the sequencing laboratory and included all samples that arrived in the laboratory within 72 h of collection[28]. The median time from sample receipt to library preparation was 2.6 d (IQR 1.8–3.6 d). Sequencing had a median duration of 1 d (IQR 1.0–1.1 d), whereas the time from sequencing completion to bioinformatics analysis and diagnostic report generation had a median of 2 d (IQR 1.0–3.9 d) (Fig. 4b).

We compared the time from sample receipt in the laboratory to generation of a diagnostic report between liquid biopsy and tissue biopsy in a paired analysis. The TAT for liquid biopsy (median: 6.5 d, IQR 5.1–10.5 d) was 40.3 d earlier than the TAT for tissue biopsy (median: 46.8 d, IQR 20.1–192.4 d) ($P = 4.42 \times 10^{-10}$) (Fig. 4c).

**Discussion**
Liquid biopsies have gained notable attention for their clinical potential across various cancer types[29], prompting efforts to harness this

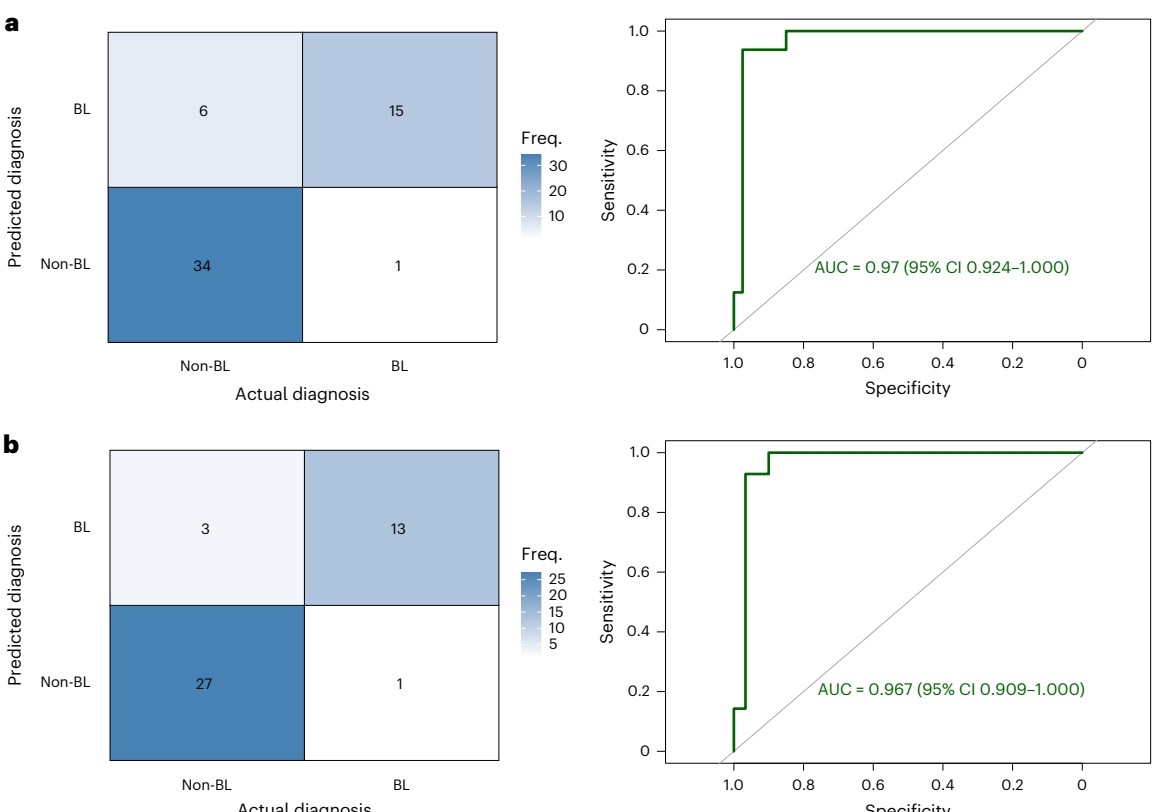

**Fig. 2 | External validation of the diagnostic model. a**, Confusion matrix (left) and ROC curve (right) for external validation using imputed data ($n = 56$). The heatmap shows agreement between predicted and actual diagnoses (BL versus non-BL) and the ROC curve summarizes overall discriminative performance (AUC with 95% CI indicated). **b**, Confusion matrix (left) and ROC curve (right) for external validation restricted to complete cases ($n = 44$). Performance metrics are shown as in **a**. Sensitivity and specificity are derived from the confusion matrices and the diagonal line in ROC plots indicates no discrimination. Freq., frequency.

technology to improve existing early and precise cancer detection[15]. However, integration into routine diagnostic practice remains limited and further evidence is needed to support clinical implementation, particularly in low-resource settings[29–31].

The aggressive nature of BL necessitates early and accurate diagnosis to enable timely treatment and maximize clinical benefit[32]. This study evaluates the performance of a minimally invasive blood-based liquid biopsy approach designed for rapid BL diagnosis in children and young people presenting with clinical features of lymphoma, classifying cases as either BL or non-BL. Diagnosis in this study relied on a limited IHC panel optimized for resource-limited settings, previously shown to have 81% concordance with SoC histopathology[12]. Although the use of a simplified panel introduces diagnostic limitations, it reflects the realities of regions where BL is most prevalent and where limited access to comprehensive histopathology compels treatment initiation based on clinical features alone[33]. Quantification of plasma EBV DNA has been shown to distinguish those with BL from healthy individuals, with sensitivity of 88% and specificity of 100%[34]. In the present study, however, the EBV quantitative model achieved lower sensitivity (57%) and specificity (95%), likely reflecting the inclusion of other EBV-associated malignancies such as HL and DLBCL rather than healthy controls. Similarly, a recent, large, multi-country study reported high diagnostic accuracy for EBV DNA quantification using digital droplet PCR, but again relied on comparisons with healthy population controls, limiting applicability to real-world clinical settings where multiple EBV-associated malignancies coexist[35]. Together, these findings suggest that, although EBV quantification is a useful screening tool, it lacks sufficient specificity for diagnostic confirmation. Incorporating additional EBV attributes alongside mutation-based and translocation-based markers therefore enhances diagnostic

specificity and biological interpretability consistent with observations in EBV-associated nasopharyngeal carcinoma[26].

Although acute EBV infection can be associated with elevated EBV copy numbers, integration of both quantitative and qualitative EBV parameters enabled reliable distinction between BL and reactive cases (Supplementary Fig. 1). Improved performance was achieved by incorporating additional liquid biopsy parameters, including *MYC*–Ig translocation, *MYC* intron 1 and exon 2 mutation counts and autosomal fragment entropy, resulting in 73% sensitivity and 95% specificity. Combining liquid biopsy features with clinical variables yielded the highest diagnostic performance (AUC of 0.98, sensitivity 0.94 and specificity 93%). Notably, model performance was primarily driven by molecular features, rather than clinical parameters (Extended Data Fig. 6), underscoring the central role of liquid biopsy markers in BL discrimination.

Due to the rational panel design, *MYC–IGH* translocations were detectable in 48% of cases of BL. Although translocation-positive cases showed marginally higher sequencing depth, median coverage across both groups exceeded 1,000× (Supplementary Table 3), making insufficient depth an unlikely explanation for undetected events. Instead, this likely reflects the heterogeneous distribution of *MYC* breakpoints in BL, many of which lie outside the targeted region. Expanding coverage could improve sensitivity but would increase cost and complexity, conflicting with the goal of a scalable, cost-efficient assay for use in low-resource settings.

Distinguishing true somatic mutations from germline SNPs remains challenging in tumor-only sequencing, particularly in African populations that are underrepresented in reference genomes and variant databases. Although matched germline DNA was unavailable, stringent filtering was applied to mitigate this limitation. Nevertheless, some intronic SNPs may have escaped detection, highlighting the

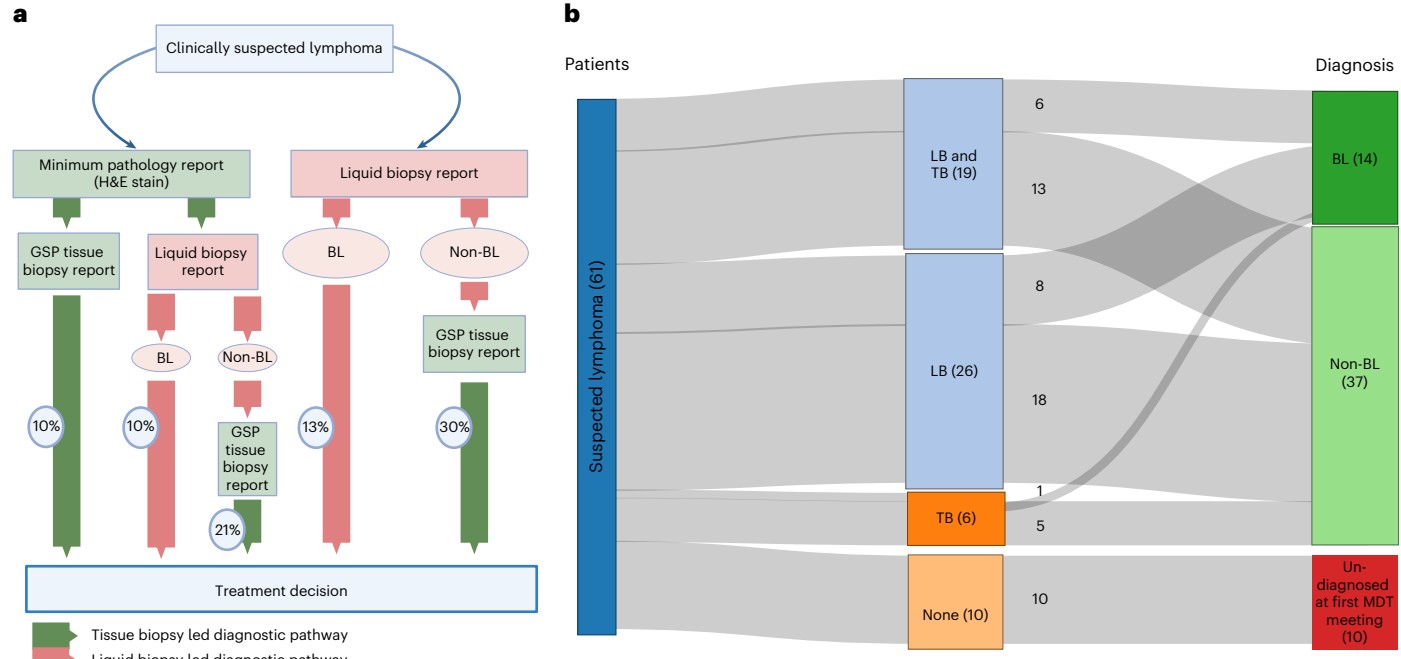

**Fig. 3 | Diagnostic pathways and test availability at the first MDT meeting.** **a**, Diagnostic and treatment decisions for children and young adults with clinically suspected lymphoma based on tissue biopsy (green) or liquid biopsy (pink) at the first MDT meeting. In 16% of cases, diagnosis was established at a subsequent MDT meeting, not shown. **b**, Sankey diagram illustrating the availability of liquid biopsy (LB) and tissue biopsy (TB) results at the first MDT meeting and the corresponding diagnostic outcomes (*n* = 61). Panel **a** created in BioRender; Claudius, C. https://biorender.com/rmkrptd (2026).

need for population-specific genomic reference datasets to support accurate interpretation of molecular findings in underrepresented African populations.

Obviously, for non-BL cases, standard pathology remains essential for accurately diagnosing other lymphoma subtypes, identifying non-lymphoid malignancies or confirming benign lesions. Although none of the acute EBV infection cases was misclassified as BL, underscoring the model's ability to distinguish malignant from reactive EBV-driven states, the number of cases of acute infection was limited. Evaluation in an even larger and more diverse cohort, incorporating additional acute EBV infection samples, would further substantiate the model's specificity and clinical applicability.

Our study demonstrates the feasibility of integrating liquid biopsy into routine clinical workflow in underresourced African hospitals using an MDT approach. Incorporation of liquid biopsy increased diagnostic yield for BL and shortened the time to diagnosis, with 93.3% of cases of BL diagnosed within 1 week, compared with 40% using tissue biopsy alone.

Although liquid biopsy achieved significantly shorter turnaround times than GSP tissue biopsy, consistent with studies from high-income countries[36,37], several limitations merit consideration. First, future service configuration (centralized or decentralized), automation and health system integration remain uncertain and may affect turnaround times at scale. Second, study conditions likely benefitted from enhanced oversight and quality control. The 10% (6 out of 62) tissue biopsy failure rate observed in phase II may underestimate rates in routine care, particularly in resource-limited settings, potentially limiting generalizability[38]. The shorter TAT of liquid biopsy compared with IHC reflects differences in workflow, staffing and infrastructure. In Tanzania, existing molecular skills among laboratory scientists enabled rapid training for cfDNA workflows, supported by automated bioinformatics and clinician-led interpretation using a predefined diagnostic algorithm.

In contrast, the longer TAT observed for tissue-based diagnosis reflects constrained histopathology capacity, with few pathologists and prolonged training requirements. Diagnostic delays are largely procedure related, driven by surgical biopsy processing time and delayed IHC interpretation due to heavy workload and occasional repeat biopsies[9].

Implementing cfDNA diagnostics in SSA will require alignment with existing clinical pathways. CfDNA testing could complement limited histopathology by providing rapid, minimally invasive diagnosis deployable from peripheral hospitals with centralized sequencing. Leveraging established molecular networks for HIV or tuberculosis testing may facilitate integration. Implementation research, including pragmatic pilots and health-economic analyses, will be essential to address scalability, cost-effectiveness and sustainability adoption within national cancer control programs.

Ongoing studies are extending this cfDNA approach to the quantitative detection of minimal residual disease in children receiving therapy for EBV⁺ BL, to determine its utility in predicting treatment response and clinical outcomes.

Finally, there are considerable costs associated with next-generation sequencing (NGS) technology. In a previously published health-economic microcosting analysis conducted by our group, the average per-patient cost of histopathology was estimated at US$185.01, driven primarily by staining (US$87.20, largely IHC consumables) and the biopsy procedure (US$72.29). In the same analysis, liquid biopsy cost $710.15 per patient at current throughput, with sequencing reagents representing the largest component. ($175.48 per sample)[39,40]. Although liquid biopsy is more expensive, improved clinical outcomes could justify the higher upfront cost. Ongoing modeling work by our group is evaluating the cost-effectiveness of earlier diagnosis using liquid biopsy, with preliminary findings supporting further evaluation in implementation studies. Adoption of additional sequencing tests for other indications across infectious and noncommunicable diseases using high-throughput sequencing platforms is likely to lead to substantial cost reduction, similar to that observed with HIV viral load or human papillomavirus testing[41,42]. Ultimately

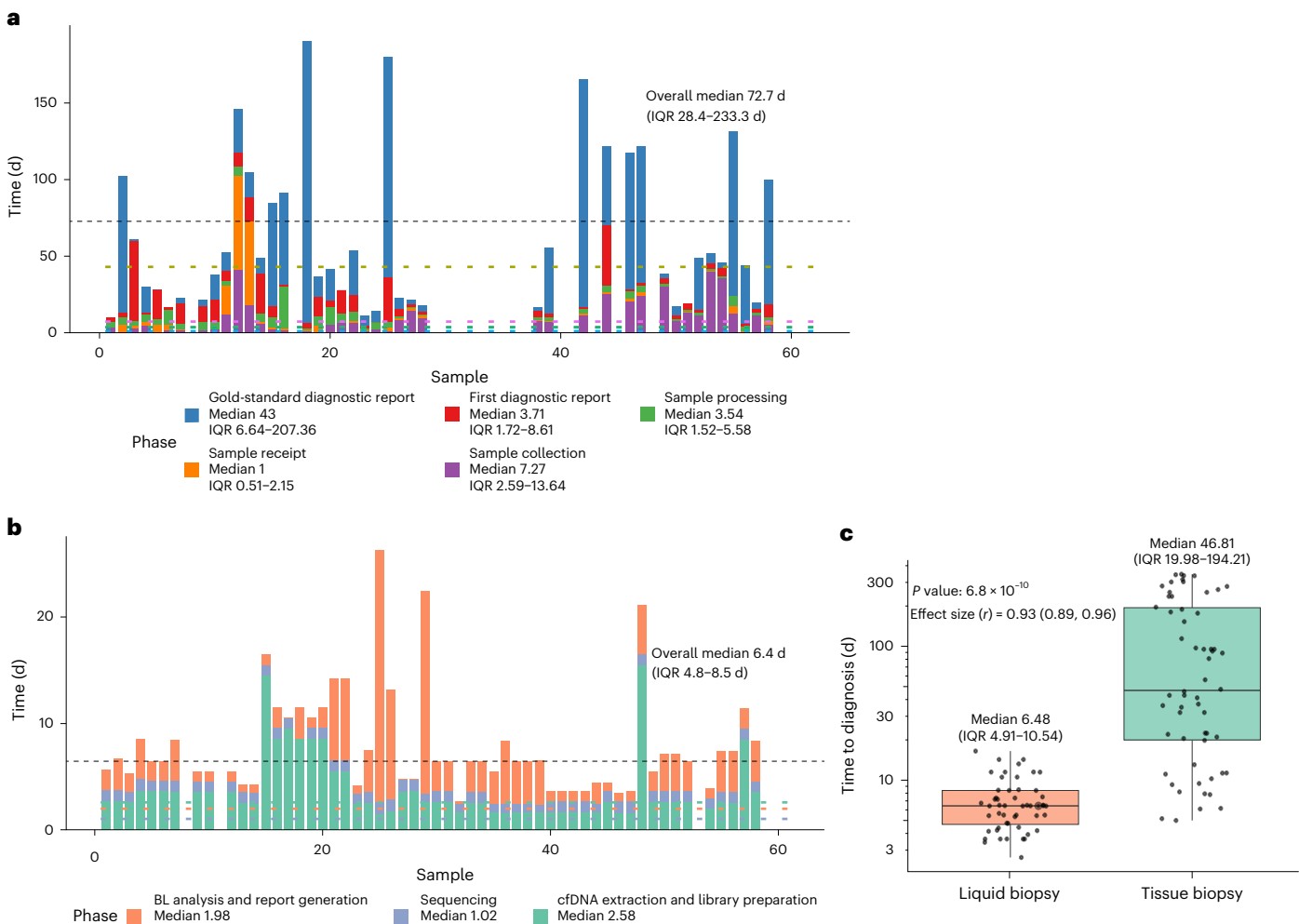

**Fig. 4 | Time from sample collection to diagnosis for tissue and liquid biopsies.** **a**, TAT for tissue biopsy from the sample collection to diagnostic reporting, shown per patient and stratified by a processing step. **b**, TAT for liquid biopsy from sample receipt to diagnostic reporting, shown per patient and stratified by a processing step. **c**, Comparison of time to diagnosis between liquid biopsy and tissue biopsy. The center lines denote medians, the boxes indicate IQRs, the whiskers extend to values within 1.5× the IQR and the points represent individual samples. Groups were compared using a two-sided, paired Wilcoxon's signed-rank test; no adjustment for multiple comparisons was applied.

this might even allow the expansion of liquid biopsy testing for BL into primary care settings in endemic regions.

## Online content

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

**Clara Chamba** [1✉], **Heavenlight Christopher**[1], **Emmanuel Josephat**[1], **Julius Sseruyange**[2], **Alisen Ayitewala**[2], **Kieran Howard** [3], **Helene Dreau**[3], **Adam Burns** [3], **Ismail Legason** [4], **Isaac Otim**[4], **Priscus Mapendo**[5], **Leah Mnango**[6], **Advera Ngaiza**[6], **Alex Mremi** [7], **Edrick Elias**[8], **Carol Achola**[2], **William Mawalla** [1], **Rehema Shungu**[1], **Eli Mkwizu**[7], **Lulu Chirande**[1], **Hadija Mwamtemi** [6], **Salama Mahawi**[6], **Godlove Sandi** [6], **Heronima J. Kashaigili**[8,9], **Sıla Gerlevik**[3], **Paul Shadrack Ntemi**[9], **Erick Magorosa**[6], **Daniel Mbwambo**[5], **Malale Tungu**[1], **Martin Ogwang**[4], **Faraja Chiwanga**[1,6], **Sam M. Mbulaiteye** [10], **Claire El Mouden**[3], **Emmanuel Balandya**[1], **Anthony Cutts**[3], **Liz Morrell** [3], **Dimitrios Vavoulis** [3] & **Anna Schuh** [1,3]

[1]Muhimbili University of Health and Allied Sciences, Dar-es-Salaam, Tanzania. [2]Central Public Health Laboratories, Kampala, Uganda. [3]University of Oxford, Oxford, UK. [4]St. Mary's Hospital Lacor, Gulu, Uganda. [5]Kilimanjaro Christian Medical Centre, Moshi, Tanzania. [6]Muhimbili National Hospital, Dar-es-Salaam, Tanzania. [7]KCMC University, Moshi, Tanzania. [8]Catholic University of Health and Allied Sciences, Mwanza, Tanzania. [9]Bugando Medical Centre, Mwanza, Tanzania. [10]National Cancer Institute, Bethesda, MD, USA. ✉e-mail: clarachamba@gmail.com

## Methods

This study was approved by the Oxford Tropical Research Ethics Committee (OxTREC: no. 15-19), the National Institute of Medical Research in Tanzania (no. NIMR/HQ/R.8a/Vol.IX/3408) and the Uganda National Council of Science and Technology (UNCST: no. HS529ES), and was conducted in accordance with the Declaration of Helsinki. Written informed consent was obtained from parents or legal guardians of all participating children, with age-appropriate assent obtained where applicable. Participants did not receive financial compensation for study participation.

### Study design

This study used data from the prospective, multicenter, observational, Aggressive Infection-Related East Africa Lymphoma (AI-REAL) study[43], which was conducted in two phases. In phase I, we trained different liquid biopsy models and assessed their performance for the diagnosis of EBV+ BL using histopathology with a limited IHC panel and review by a minimum of 2 histopathologists as the GSP in 212 participants with clinically suspected lymphoma. In phase II, we performed a head-to-head comparison between the liquid biopsy and gold-standard tissue biopsy to assess the TAT for the two methods ($n = 58$ participants with clinically suspected lymphoma) and to conduct an additional independent external validation for the best-performing model ($n = 56$ participants with clinically suspected lymphoma).

### Participants

The study enrolled all children and young adults, from the age of 3 years to 25 years, clinically suspected of having lymphoma and consenting to participate in the study from August 2019 to July 2023 from four hospitals in Tanzania and Uganda, that is, the Muhimbili National Hospital, Kilimanjaro Christian Medical Centre, Bugando Medical Centre and St. Mary's Hospital Lacor. Participants with suspected lymphoma who had previously received chemotherapy, immunotherapy or any investigational agent for lymphoma, or participated in a clinical trial on lymphoma, were excluded. Phase I of the study was conducted from August 2019 to July 2022 and enrolled a total of 313 participants, whereas phase II was conducted from August 2022 to July 2023 and enrolled a total of 64 participants (Extended Data Fig. 1).

Tissue biopsies were obtained by a qualified surgeon through excisional biopsy and immediately fixed in 10% neutral buffered formalin. The formalin-fixed specimens were then transported to the local pathology laboratory for further processing and diagnostic evaluation. In parallel, liquid biopsy samples were collected via venipuncture into Roche Cell-Free DNA Collection Tubes, following standard phlebotomy procedures. The tubes were transported to the molecular laboratory, where plasma was separated by centrifugation and stored at −80 °C until further analysis.

### Laboratory procedures

Extended Data Fig. 7 provides a summarized overview of the laboratory procedures performed for both the liquid and the tissue biopsies. All measurements were taken from distinct, independent samples.

**Gold-standard pathology.** Before study initiation, we performed a comprehensive training and infrastructure needs assessment at the three local study pathology laboratories. To ensure optimal histopathology assessment for the study and local capacity building, technical staff and the four clinical study pathologists underwent training, followed by competency assessments conducted by expert hematopathologists in tissue fixation, embedding, sectioning and IHC staining techniques. The training included hands-on sessions and on-site mentoring over a period of 1 year. An automated IHC staining platform (Ventana Benchmark GX) was installed and maintained at all participating laboratories for the duration of the study and 3-monthly internal quality control procedures were implemented, including the use of known positive controls. Tissue samples were fixed in 10% neutral buffered formalin and embedded in paraffin according to standard protocols. H&E staining was performed on all samples to evaluate morphology and IHC was done using the previously described limited IHC antibody panel for BL.

This diagnostic approach utilizes a three-phase scoring system. The first phase combines typical BL morphology with immunostains BCL2, CD10 and CD20 to establish a diagnosis. Cases that remain inconclusive proceed to a second phase with CD38, CD44 and Ki67, whereas unresolved cases undergo a third phase with FISH analysis for *MYC*–Ig translocation. In our study, we used the first-phase immunostains (BCL2, CD10 and CD20) to support the diagnosis of BL, because this phase has previously been shown to achieve 81% concordance with the World Health Organization SoC pathology and is well suited for the limited-resource settings[12,44]. During the study, we purchased and procured all relevant antibodies and ensured that these were available. Finally, we introduced digital whole-slide imaging to enable slide review by a minimum of two local, study histopathologists and to gauge external third opinions as required[45]. This approach was adopted as the gold standard for evaluating the performance of the liquid biopsy methods.

**CfDNA extraction, library preparation and targeted sequencing.** CfDNA extraction, library preparation and targeted sequencing were performed at two dedicated study laboratories; the Haematology clinical Research Laboratory at the Muhimbili University of Health and Allied Sciences in Tanzania and the Central Public Health Laboratory.

*Sample collection and processing.* Approximately 8 ml of whole blood was collected in Roche circulating cfDNA (ccfDNA) tubes or PAXgene blood ccfDNA tubes. Plasma was separated by centrifugation at 1,600$g$ for a duration of 15 min continuously, followed by a second centrifugation of the separated plasma at 4,500$g$ for 15 min. Separated plasma was stored in 1.5-ml Eppendorf tubes at −80 °C.

*Extraction and quantification of cfDNA.* CfDNA was extracted from plasma using the QIAamp Circulating Nucleic Acid Kit (QIAGEN, cat. nos. 51304 and 51306) according to the manufacturer's instructions. Briefly, it involved four main steps; first, samples are lyzed to inactivate DNases and RNases and allow complete release of nucleic acids from bound proteins, lipids and vesicles. Second, the lysates are transferred onto a QIAamp Mini column and circulating nucleic acids are adsorbed from a large volume on to the small silica membrane as the lysate is drawn through by vacuum pressure. Third, while the nucleic acids remain bound to the membrane, contaminants are washed away in a three-step wash process. The final step involves elution of highly purified nucleic acid. The quantification of cfDNA was done with a Qubit fluorimeter 3.0 using the qubit high-sensitivity assay (Thermo Fisher Scientific).

*Library preparation and targeted sequencing.* Libraries were constructed using 50 ng (in 30 μl) of extracted cfDNA with the ThruPLEX Tag-Seq HV kit according to the manufacturer's protocol. The process involved addition of unique ThruPLEX HV Unique Dual indexed PCR primers to aid with sample tracking. Seven cycles were performed to ensure a yield of >500 ng depending on the concentration of the input DNA. This was followed by a purification step through magnetic separation using AMPure XP beads (Beckman Coulter). The final library was quantified and validated using the Qubit HS kit and the Bio-analyser High Sensitivity DNA kit, according to the manufacturer's instructions. Library hybridization and capture were done using the xGen hybridization capture kit according to the manufacturer's instructions.

*Panel design and verification.* The procedure used a custom-made NGS panel targeting mutational hotspots in 17 EBL-related and HL-related genes (*MYC, IGH, IGK, IGL, ID3, TP53, TNFAIP3, NFKBIE, SOCS1, EP300,*

*BTK*, *STAT6*, *CSF2RB*, *ITPBK*, *XPO1* and *B2M*), selected based on results from previous studies and EBV genes expressed in latently infected cells: *EBER1*, *EBER2* and *EBNA2*. The final panel manufactured by Integrated DNA Technologies (IDT) consisted of 731 probes at a size of 148 kb to permit compatibility with either the iSeq100 or the MiSeq sequencing platforms. The final panel design with the genomic coordinates of our genes of interest is listed in Supplementary Table 4. Target enrichment was performed using a hybrid capture approach and sequencing was conducted twice weekly on the MiSeq platform using the MiSeq reagent kit v2 (300 cycles) at a loading concentration of 10 pM, with 6 samples per run.

*Variant calling*. Structural variants were detected by split-read analysis using GRIDSS and SNVs were called using VarScan 2[46]. The following filtering strategies were used to call somatic variants:

(1) Panel of normal plasmas: we excluded variants that occurred in ≥2 samples in a separate cohort of 12 healthy controls.

(2) Population databases: all variants were annotated against gnomAD and those with a population allele frequency (minimum allele frequency (MAF)) > 1% were excluded.

(3) VAF: to minimize the possibility of including germline variants, we excluded all variants with VAF ≥ 40%, in addition to the initial exclusion of low-level artefacts with VAF < 1%.

(4) Sequencing depth and read support: only variants with a minimum sequencing depth of 500× and at least 5 mutant reads were retained for downstream analysis.

(5) Copy number alterations (CNAs): we excluded variants that showed evidence of being affected by CNAs. This was done through an internal normalization approach in which the expected relationship between VAF and $\log_{10}$(transformed ctDNA) was modeled using *ID3* mutations (rarely affected by CNAs), and the resulting tolerance limits (±2× median absolute deviation of residuals) were applied to *MYC* and *TP53* variants to flag likely CNA-affected outliers (Extended Data Fig. 4a,b and Supplementary Table 6).

Samples that had GSP tissue biopsy results were prioritized for sequencing to allow for clinical validation. For phase II samples, batching was not applied; instead, samples were transported to the sequencing laboratory immediately after collection and sequenced in a head-to-head comparison against GSP tissue biopsy, allowing us to measure TAT prospectively. The same twice-weekly sequencing schedule was maintained, with each MiSeq run limited to a maximum of six samples. Details on the target coordinates (Supplementary Table 4) and sequencing parameters (Supplementary Table 5 and Supplementary Fig. 2) are described in Supplementary Information.

**Bioinformatics analysis**
The sequence data were analyzed and stored by a bespoke pipeline (courtesy of the Oxford Molecular Diagnostic Centre) in a Health Insurance Portability and Accountability Act of 1996-compliant, Amazon Web Services cloud. Paired reads were combined and statistics generated on the total number of reads and invalid reads, using a customized tool (udini). The raw sequence data in FASTQ format were aligned to GRCh37 (hs37d5) with BWA-MEM2. Sorting was done using samtools. De-duplication of reads and generation of error rates and family sizes were done by a custom-made tool (elduderino). The de-duplicated FASQ was re-aligned to GRCh37 with BWA-MEM2 and fragment sizes calculated using read pairs via a customized script (available on request). Another customized script (available on request) was used to map the reads on to the genomic regions (targeted genes) of interest and to calculate copies per cell for the EBV genes (mean depth of EBV gene per mean depth for all other targeted genes). The last base at the end of each read was trimmed using a customized tool (trim) and the trimmed

SAM file converted into a BAM file and indexed. A customized script (available on request) was used to generate summary statistics for coverage of all targets at different coverage depths and for different genes.

Customized scripts (available on request) were developed for variant calling using VarScan and annotation of the variants using Ensembl Variant Effect Predictor. IgCaller (v1.2 software utilizing the hg19 reference genome) was used to comprehensively analyze the rearrangements of the Ig genes and identify oncogenic translocations and the Genomic Rearrangement Identification Software Suite, with additional Picards options and Samtools as aligner (GRIDSS, v2.13.2), was also included as a structural variation caller. To visualize and distinguish true variants from false variants, Integrated Genome Viewer (IGV, v2.13.0) was used. EBV DNA size ratio was calculated as the proportion of EBV DNA fragments with size between 180 bp and 200 bp ((Total EBV DNA with size 180–200 bp)/(Total autosomal DNA with size between 180 bp and 200 bp)). Calculation of the size ratio and selection of the fragment size window were determined based on the methodology of a previous study looking at EBV DNA in nasopharyngeal carcinoma[26]. The lower the EBV size ratio, the lower the proportion of EBV DNA molecules of size 180–200 bp. The distribution of the fragment sizes for reads that map to regions of EBV and for reads that map to the autosomes was calculated and recorded as the EBV entropy and autosome entropy, respectively. This gives a measure of how wide or clustered the distribution of fragment sizes is. The EBV DNA size ratio, EBV entropy and autosome entropy were calculated using customized scripts (Fig. 3 and Supplementary Information).

**Liquid biopsy predictor description**
The following covariates were tested as predictors in the liquid biopsy diagnostic model:

- Median VAF: calculated as the median VAF of the likely somatic mutations in *MYC*, *TP53* and *ID3*.

- CtDNA: calculated as a product of the cfDNA in hGE ml$^{-1}$ and the median VAF of likely somatic mutations.

- *MYC* intron 1 mutations: absolute number of mutations in *MYC* intron 1.

- *MYC* exon 2 mutations: absolute number of mutations in *MYC* exon 2.

- *EBER1*, *EBER2* and *EBNA2* (EBV genes) quantity (copies per cell): calculated as the mean depth of the respective EBV gene divided by the mean depth for all other targeted genes and expressed as copies per cell.

- Maximum value of EBV copies per cell (EBV$_{max}$): defined as the highest copies-per-cell value among the three EBV genes measured (*EBER1*, *EBER2* and *EBNA2*) for each sample.

- EBV fragment size ratio (EBVSR): EBV DNA size ratio was calculated as the proportion of EBV DNA fragments with size between 180 bp and 200 bp ((Total EBV DNA with size 180–200 bp)/(Total autosomal DNA with size between 180 bp and 200 bp)). Calculation of EBVSR and selection of the fragment size window was determined based on the methodology of a previous study looking at EBV DNA in nasopharyngeal carcinoma[26].

- EBVP: represents the number of EBV DNA reads divided by the total number of sequenced reads (after removal of PCR duplicates), expressed as a proportion[26].

- EBV DNA and autosomal entropy: calculated as a measure of the diversity of fragment length distributions, using Shannon entropy computed across the full distribution of paired-end sequencing fragment sizes mapping to the EBV genome (for EBV entropy) and the autosomal regions (for autosomal entropy), respectively. Fragment sizes were binned into 20-bp intervals and the proportional frequency of fragments in each bin was used to derive entropy, reflecting the overall variability and dispersion of the fragment size distribution.

## MDT meeting decision tree

To further evaluate the potential clinical utility of the liquid biopsy diagnostic test, we established a weekly virtual MDT meeting consisting of pediatric hematologists and oncologists, hematologists, pathologists and laboratory scientists from all study sites, as well as a senior study bioinformatician.

When tissue biopsy was reported based on H&E stain only (minimum pathology report) and the liquid biopsy result indicated BL, treatment was initiated as BL (Fig. 3). When liquid biopsy indicated non-BL, treatment was deferred until GSP tissue biopsy results were available. If the liquid biopsy report was unavailable at the first MDT meeting, treatment decisions were deferred until GSP tissue biopsy results were available.

In cases where BL diagnosis was definitive by liquid biopsy, treatment was initiated (Fig. 3). When liquid biopsy indicated non-BL, treatment was deferred until the GSP tissue biopsy result was available. If neither tissue biopsy nor liquid biopsy results were available at the first MDT meeting, treatment decisions were deferred until the second MDT meeting in the following week. Our analysis focused on the initial MDT review to capture the immediate outcome of the two tests in the early decision-making process for cases of BL, although enhanced follow-up protocols in phase II ensured that results from both liquid biopsy and tissue biopsy were obtained for all cases and integrated into patient management plans.

## Turnaround time

For the tissue biopsy TAT analysis, we assessed the time from patient presentation at the hospital to the issuance of the GSP diagnostic report, divided into key intervals: time from presentation to sample collection; collection to laboratory receipt; receipt to H&E slide processing; time to issuing the first diagnostic report (based on H&E review); and time to completing the final GSP diagnostic report. For the liquid biopsy TAT, we measured the time from sample receipt in the laboratory to issuance of the diagnostic report, broken down into the intervals of cfDNA extraction and library preparation, targeted sequencing, bioinformatic analysis and final report generation. For the direct TAT comparison between GSP tissue biopsy and liquid biopsy, we specifically analyzed the time from laboratory sample receipt at the respective laboratories to the issuance of the diagnostic report for each method.

## Diagnostic yield

We assessed the diagnostic yield for BL at the first MDT meeting, expressed as a percentage and calculated from the number of BL cases diagnosed at the first MDT meeting by either the GSP tissue biopsy or the liquid biopsy method divided by the total number of confirmed cases of BL.

## Statistical analysis

All statistical analyses were conducted in R v4.2.3. Normality of continuous variables was assessed using the Shapiro–Wilk test and visual inspection of histograms and Q–Q plots. As the data were not normally distributed, nonparametric summaries and tests were used where appropriate. Descriptive statistics were summarized as proportions for categorical variables or medians with IQRs for continuous variables. The Pearson's $\chi^2$ test and Wilcoxon's rank-sum test were used to assess associations between variables. We applied the Benjamini–Hochberg procedure to adjust for multiple comparisons across 19 variables. False recovery rate-adjusted $P$ values are presented and values <0.05 were considered statistically significant. All variables showing a significant association with the diagnosis of BL were considered for inclusion in a penalized logistic regression model that we previously described and validated using a smaller cohort[20]. Before model fitting, candidate predictors were examined for multicollinearity using pairwise correlation and variance inflation factor analysis and highly collinear variables

were excluded to improve model stability and interpretability. Priority was given for those that were most biologically or clinically relevant. Six different models were created based on a combination of different variables, as shown in Table 2.

For each diagnostic model, the optimal penalty parameter ($\lambda$) was selected via tenfold crossvalidation to minimize classification error and ROC curves were generated. The best-performing model was identified based on the highest AUC and pairwise comparisons of AUCs were conducted using DeLong's test to assess the statistical significance of performance differences between models. The relative importance of variables contributing to the performance of the comprehensive model was derived from the absolute values of the coefficients obtained through LASSO regression. In this penalized logistic regression framework, variables with larger absolute coefficients exert a greater influence on the model's predictions, reflecting their relative contribution to distinguishing cases of BL from non-BL cases. External validation was done by predicting the diagnosis on a different cohort (phase II) based on the best-performing model. In this analysis, missing data for LDH in 12 samples was imputed using multiple imputation by chained equations, implemented in the mice R package (v3.16.0). Predictive mean matching (method = 'pmm') was applied. Comparison of the median TAT between the liquid biopsy and the tissue biopsy was done using Wilcoxon's signed-rank test (for paired samples).

## Statistics and reproducibility

A formal sample size calculation was conducted to ensure sufficient power for assessing diagnostic accuracy. Using a binomial proportion approach ($z = 1.96$, expected sensitivity = 0.8, margin of error = 0.10), a minimum of 62 cases with BL was estimated, corresponding to approximately 124 participants, assuming a 1:1 case–control ratio. For multivariable model development, an events-per-variable threshold of 20, with up to 18 candidate predictors, indicated a target sample size of 720 participants. Owing to the prospective design and disruptions related to the COVID-19 pandemic, this target was not reached. The final cohort comprised 212 participants, including 81 cases of BL. To mitigate overfitting in the context of a constrained sample size and multiple candidate predictors, LASSO regression was used for variable selection and regularization. Model performance was internally validated using tenfold crossvalidation to assess robustness and generalizability, in accordance with recommended best practices for predictive model development.

Participants were enrolled consecutively as part of routine clinical care. No randomization was performed and investigators were not blinded to allocation during experiments or outcome assessment. Participants with missing outcome data were excluded from the analysis; this exclusion was predefined to preserve the integrity of outcome-based comparisons and was limited to individuals lacking key clinical endpoints or follow-up information required for the primary analyses. For other missing data, including covariates, appropriate imputation methods were applied to minimize bias and retain statistical power.

Reproducibility was supported through the use of standardized laboratory protocols, predefined diagnostic algorithms established a priori and automated bioinformatics pipelines applied consistently across all samples. Source data and analysis code are made available as described in 'Data availability'.

## Inclusion and ethics

This research was conducted through equitable collaboration between institutions in Tanzania, Uganda and the UK, with a deliberate focus on capacity strengthening, technology transfer and shared scientific leadership. NGS platforms were transferred and implemented locally, enabling all cfDNA sequencing to be performed in-country. Laboratory scientists, clinicians and bioinformaticians participated in structured, crosscountry training and reciprocal site visits, supporting hands-on

skills development in library preparation, sequencing, bioinformatics analysis and clinical interpretation. Bioinformatics pipelines and reporting frameworks were deployed with local oversight, ensuring full access to data and analytical workflows for all collaborating investigators. The program supported formal academic training, including PhD, DPhil and masters training for local scientists, contributing to sustainable research capacity beyond the duration of the study. Study design, implementation, data interpretation and authorship reflected shared leadership and local ownership.

### Reporting summary

Further information on research design is available in the Nature Portfolio Reporting Summary linked to this article.

## Data availability

The raw sequencing data and individual-level clinical data generated in this study cannot be made publicly available owing to ethical and data protection restrictions, because they include information from human participants collected under institutional approvals that do not permit open public deposition. Data access is governed by the study consortium and collaborating centers in Tanzania and Uganda. Requests to access the underlying anonymized data will be reviewed by the consortium's data access committee in consultation with all participating institutions. If a request is deemed scientifically sound and compliant with applicable institutional, national and international data protection regulations, de-identified and anonymized data will be shared after the execution of a data transfer agreement. Processed and anonymized data supporting the findings of this study are available in the GitHub repository: https://github.com/ClaraClaudius/CLINICAL-VALIDATION-OF-LIQUID-BIOPSY-FOR-FASTER-DIAGNOSIS-OF-EBV-POSITIVE-BURKITT-LYMPHOMA.git. This repository is provided to ensure transparency and reproducibility. All data-sharing requests should be addressed to the corresponding author (clarachamba@gmail.com). Data access requests will be reviewed within 8 weeks subject to institutional, national and crossborder regulatory requirements. Source data are provided with this paper.

## Code availability

All code used for statistical analysis and figure generation in this study are available at the following GitHub repository: https://github.com/ClaraClaudius/CLINICAL-VALIDATION-OF-LIQUID-BIOPSY-FOR-FASTER-DIAGNOSIS-OF-EBV-POSITIVE-BURKITT-LYMPHOMA.git. This repository contains fully annotated R scripts used to reproduce all main analyses, figures and tables reported in the paper.

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

## Acknowledgements

We thank the members of the AI-REAL Scientific Advisory Board—S.M.M., M. Du, R. Siebert, D. Kurtz, C. Oakes, D. Bentley, S. Gopal, D. Lo, K. Naresh and L. Leoncini—for providing their expert guidance throughout the implementation of the study and for reviewing the final draft of the manuscript. We also thank T. Scanlan, O. Henke and K. Schroeder for their mentorship role and guidance throughout the course of this study. This work was supported by the National Institute for Health and Care Research (NIHR) using UK International Development funding from the UK Government to support global health research (NIHR-RIGHT award no. 200133 AS) and support from the Intramural Research Program (support to S.M.M.), National Cancer Institute, National Institutes of Health, US Department of Health and Human Services. The funder had no role in study design, data collection and analysis, decision to publish or preparation of the manuscript. The content of this publication does not necessarily reflect the views or policies of the Department of Health and Human Services, nor does mention of trade names, commercial products, or organizations imply endorsement by the U.S. Government.

## Author contributions

The study was led by A.S., F.C., C.C., M.O. and C.E.M. S.M., G.S., H.M., L.C., P.N., E.M. and I.O. contributed to the collection of clinical data at the different sites. Liquid biopsy wet lab analysis was conducted by P.M., H.C., E.J., J.S., A.A., I.L., I.O., A.B., H.D. and A.C. D.M. and E.M. processed the tissue biopsy samples and histopathology evaluation of samples was done by L.M., A.N., E.E., A.M. and C.A. W.M. managed the data for this study on the redcap database. K.H. performed the bioinformatics analysis and R.S., C.C., S.G., K.H. and D.V. conducted the statistical analysis. M.T. and L.M. contributed to the cost-effectiveness aspect of the study. C.C., A.S., S.M.M., D.V. and E.B. wrote the manuscript with contributions from all authors.

## Competing interests

The authors declare no competing interests.

## Additional information

**Extended data** is available for this paper at https://doi.org/10.1038/s41591-026-04291-z.

**Correspondence and requests for materials** should be addressed to Clara Chamba.

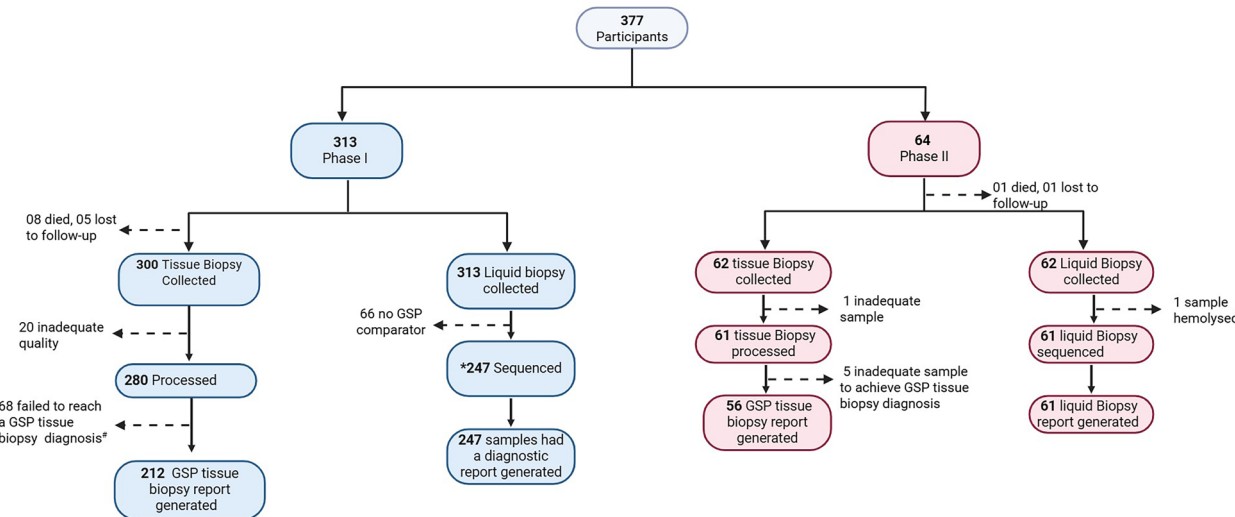

GSP = Gold Standard Pathology (Immunohistochemistry with limited panel (Naresh et al) and reviewed by at least 2 histo-pathologists.
*35 samples were still sequenced despite absence of GSP comparator

**Extended Data Fig. 1 | Study flow diagram for Phase I and Phase II cohorts.**
Flowchart showing enrolment, sample collection, processing and diagnostic outcomes for 377 participants. In Phase I (n = 313), tissue biopsy and liquid biopsy samples were collected, processed and sequenced, with generation of gold standard pathology (GSP) tissue biopsy reports and liquid biopsy diagnostic reports, including losses due to death, loss to follow-up and inadequate sample quality. In Phase II (n = 64), corresponding tissue and liquid biopsy workflows are shown, including exclusions due to inadequate or hemolyzed samples. Numbers indicate samples at each step. GSP denotes gold standard pathology.

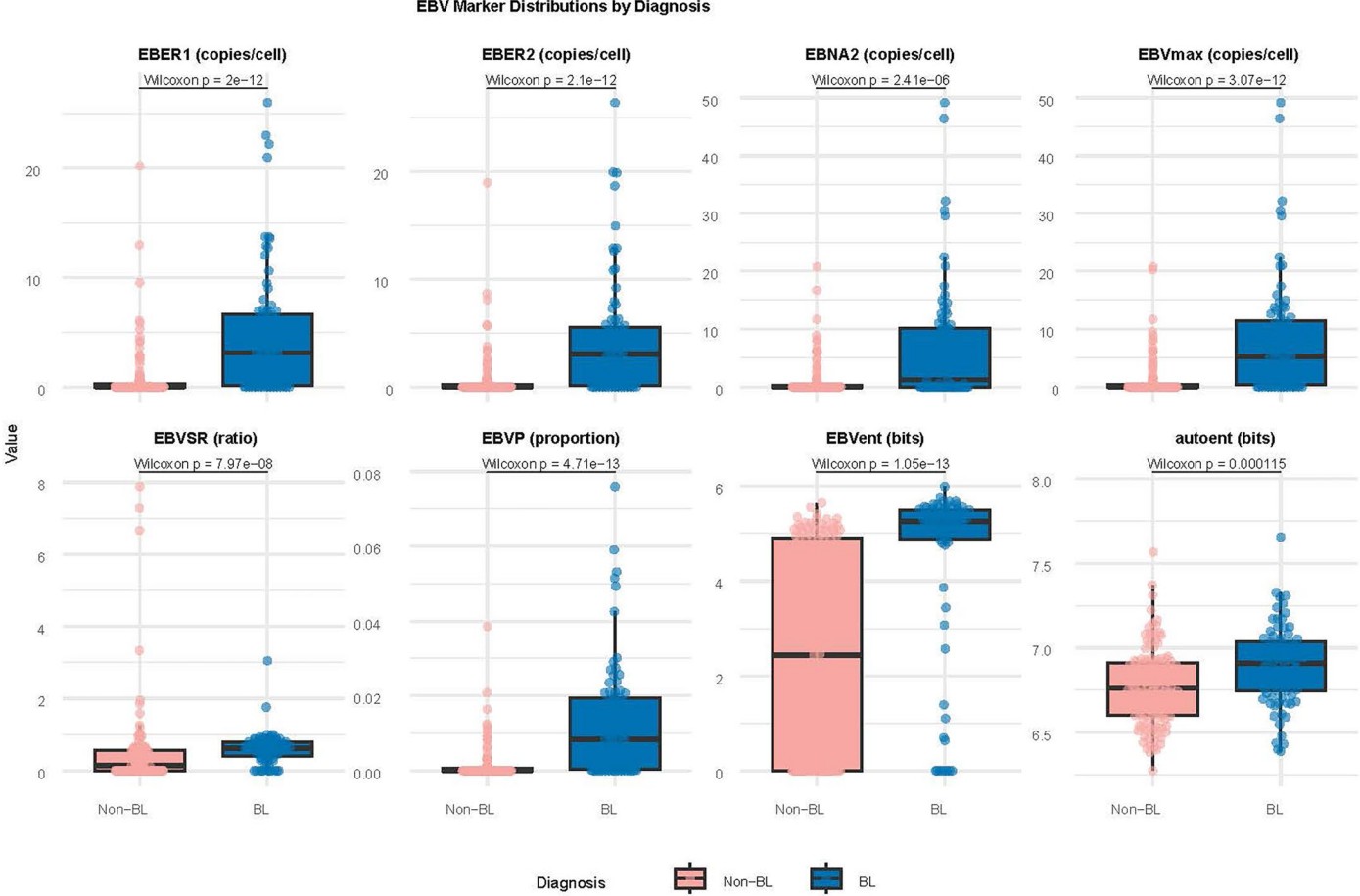

**Extended Data Fig. 2 | Distribution of EBV-derived molecular markers by diagnosis.** Box-and-whisker plots show EBV markers in BL and non-BL cases, including EBER1, EBER2, EBNA2, EBVmax (copies per cell), EBVSR (ratio), EBVP (proportion), EBV entropy (EBVent), and autosomal fragment entropy (autoent). Centre lines indicate medians; boxes show the interquartile range (IQR); whiskers extend to the minimum and maximum values within 1.5×IQR; points represent individual samples. Group comparisons were performed using two-sided Wilcoxon rank-sum tests, with P values shown for each marker (n = 212 samples).

## MYC GENE

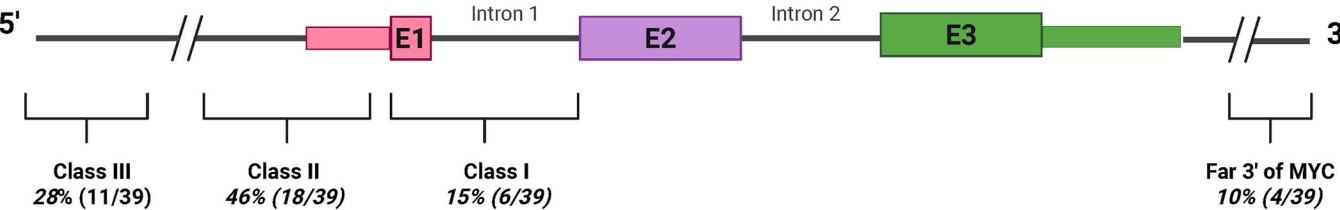

## IGH GENE

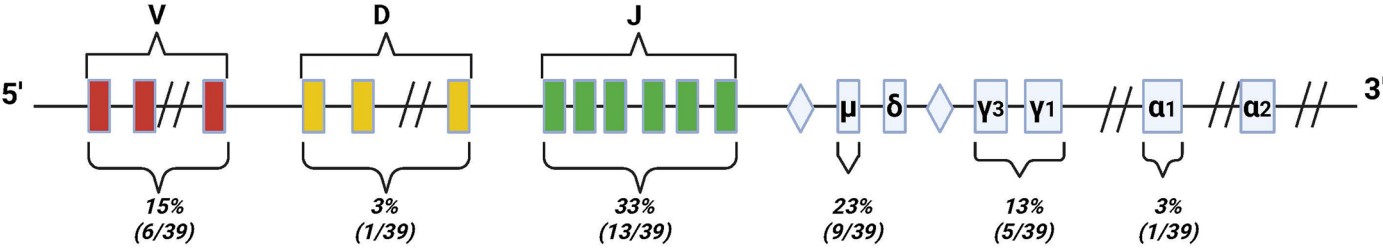

**IGK : 8% (3/39), IGL : 3% (1/39)**

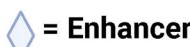

**E : Exon**

◇ **= Enhancer**

**Extended Data Fig. 3 | Genomic distribution of *MYC*–immunoglobulin translocation breakpoints.** Schematic representation of breakpoint locations across the *MYC* gene (top) and immunoglobulin loci (bottom) in Burkitt lymphoma (n = 39). Percentages indicate the proportion of cases with breakpoints in each annotated region. *MYC* breakpoints are distributed across exon 1 (E1), intron 1, exon 2 (E2), exon 3 (E3) and regions distal to the 3′ end, corresponding to class I–III translocations. *IGH* breakpoints localize to V, D and J segments and constant regions, with additional translocations involving *IGK* and *IGL*. Diamonds denote enhancer elements; boxes denote exons.

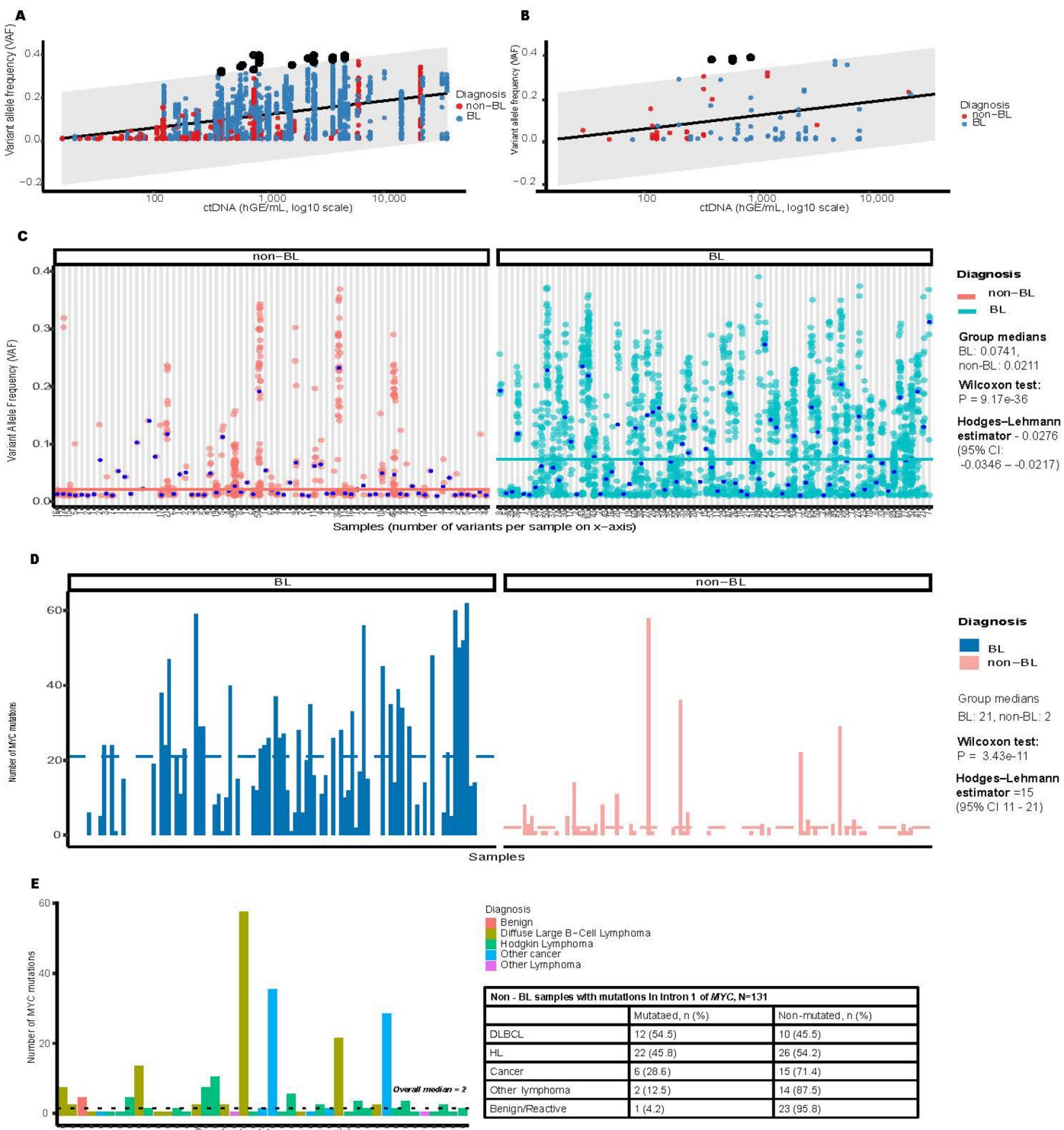

**Extended Data Fig. 4 | *MYC* and *TP53* variant allele frequency (VAF) patterns and *MYC* intron 1 mutation burden in BL and non-BL samples. A,B,** ctDNA–VAF relationships for *MYC* (A) and *TP53* (B) variants. The solid line shows the expected VAF–ctDNA relationship derived from copy-neutral *ID3* variants; the shaded band indicates the expected range. Variants outlined in black fall outside this range and are flagged as copy number alteration (CNA)–affected (n = 147 samples). **C,** Distribution of VAFs for *MYC, TP53* and *ID3* variants per sample, stratified by diagnosis. Each dot represents an individual variant; blue dots indicate sample medians and solid lines denote group medians. Differences in median between

groups was assessed with values calculated using a two-sided Wilcoxon rank-sum test. Hodges–Lehmann estimator (HL) is shown with 95% confidence intervals. Only samples with detectable variants are shown (n = 145; non-BL = 71, BL = 74). **D,** Number of *MYC* intron 1 mutations per sample in BL and non-BL groups. Bars represent individual samples; dashed lines indicate group medians. Only samples with *MYC* intron 1 mutations are shown (n = 110). **E,** *MYC* intron 1 mutation counts in non-BL samples stratified by diagnosis, with the accompanying table showing proportions of mutated and non-mutated samples. Source Data are provided as source data 4A, 4B, 4C, 4D and 4E files.

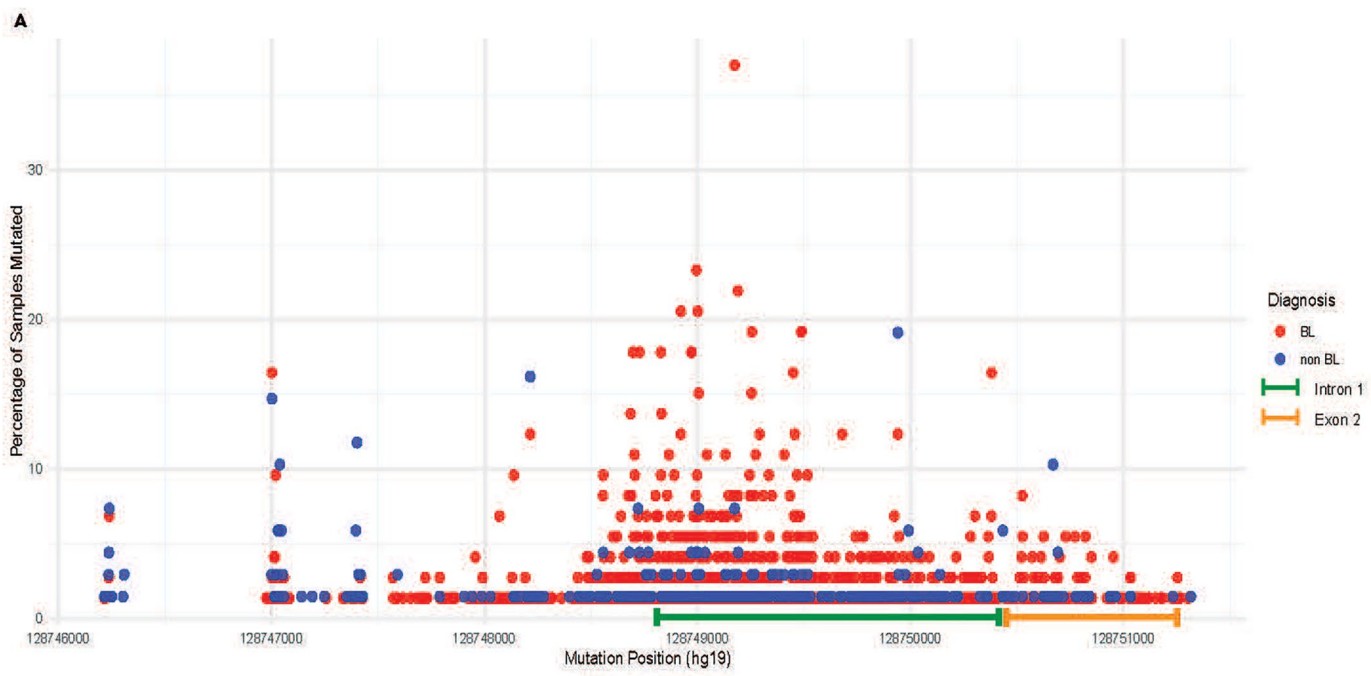

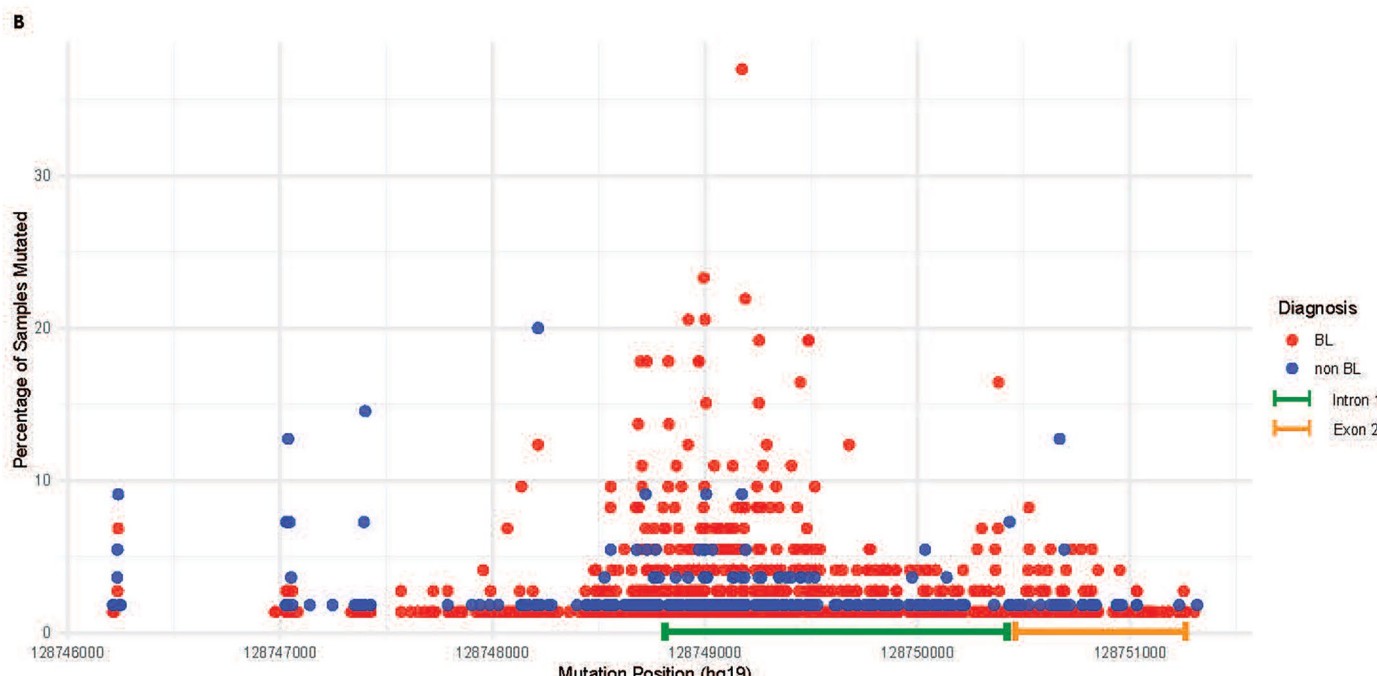

**Extended Data Fig. 5 | Distribution of *MYC* mutations in BL and non-BL samples (n = 141, BL: 73, non-BL: 68). A**, Proportion of BL and non-BL samples harbouring *MYC* mutations, expressed as a percentage of samples. **B**, Distribution of *MYC* mutations after exclusion of variants located within homopolymer regions (≥ 5 consecutive identical bases).

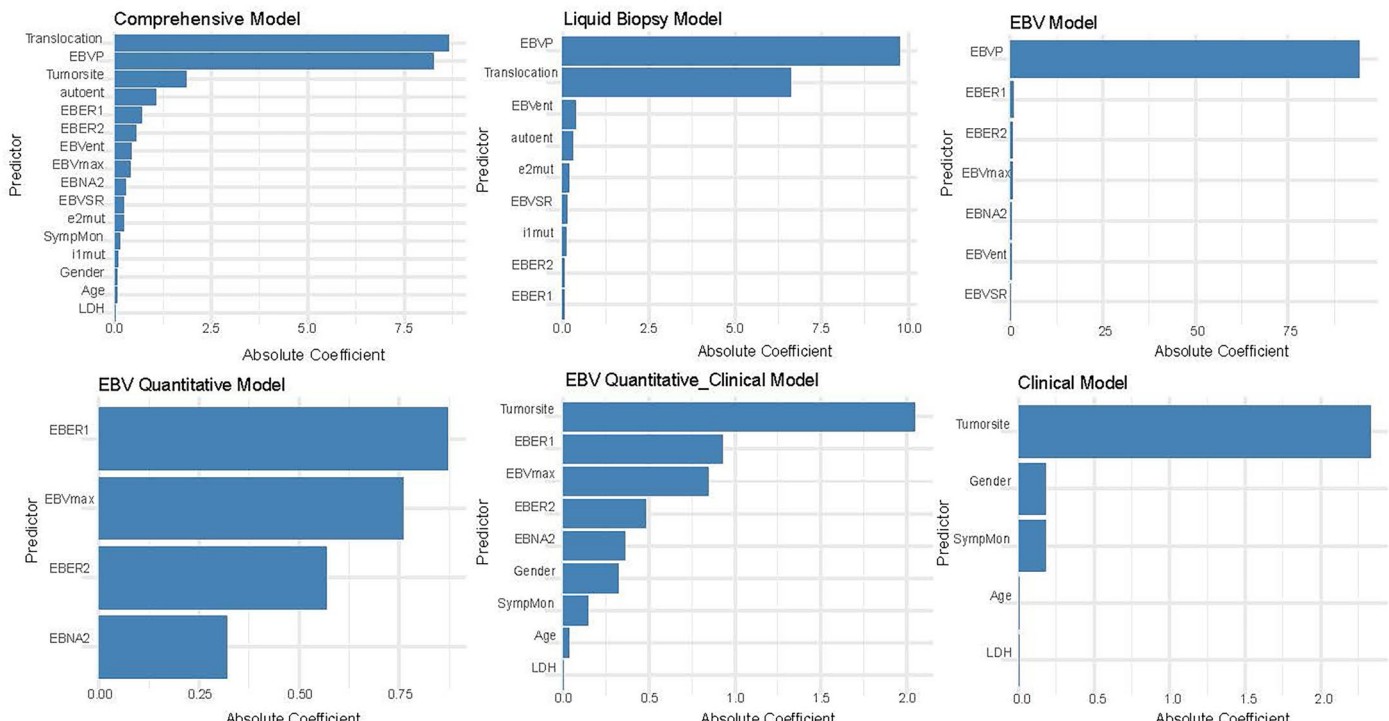

**Extended Data Fig. 6 | Importance of key predictors of Burkitt lymphoma identified by LASSO models, n = 212 samples.** Bar plots show predictors ranked by absolute coefficient magnitude from LASSO-regularised logistic regression models. Models include: comprehensive (all variables), liquid biopsy only, EBV markers only, EBV quantitative markers only, EBV quantitative plus clinical variables, and clinical variables only. Larger absolute coefficients indicate stronger contributions to classification. EBV, Epstein–Barr virus; EBVP, EBV proportion; EBVmax, maximum EBV copy number; EBER1/2, EBV-encoded RNA 1/2; EBNA2, EBV nuclear antigen 2; EBVSR, EBV size ratio; EBVent, EBV fragment entropy; autoent, autosomal fragment entropy; Tumorsite, anatomical tumour site; SympMon, symptom duration in months; LDH, lactate dehydrogenase.

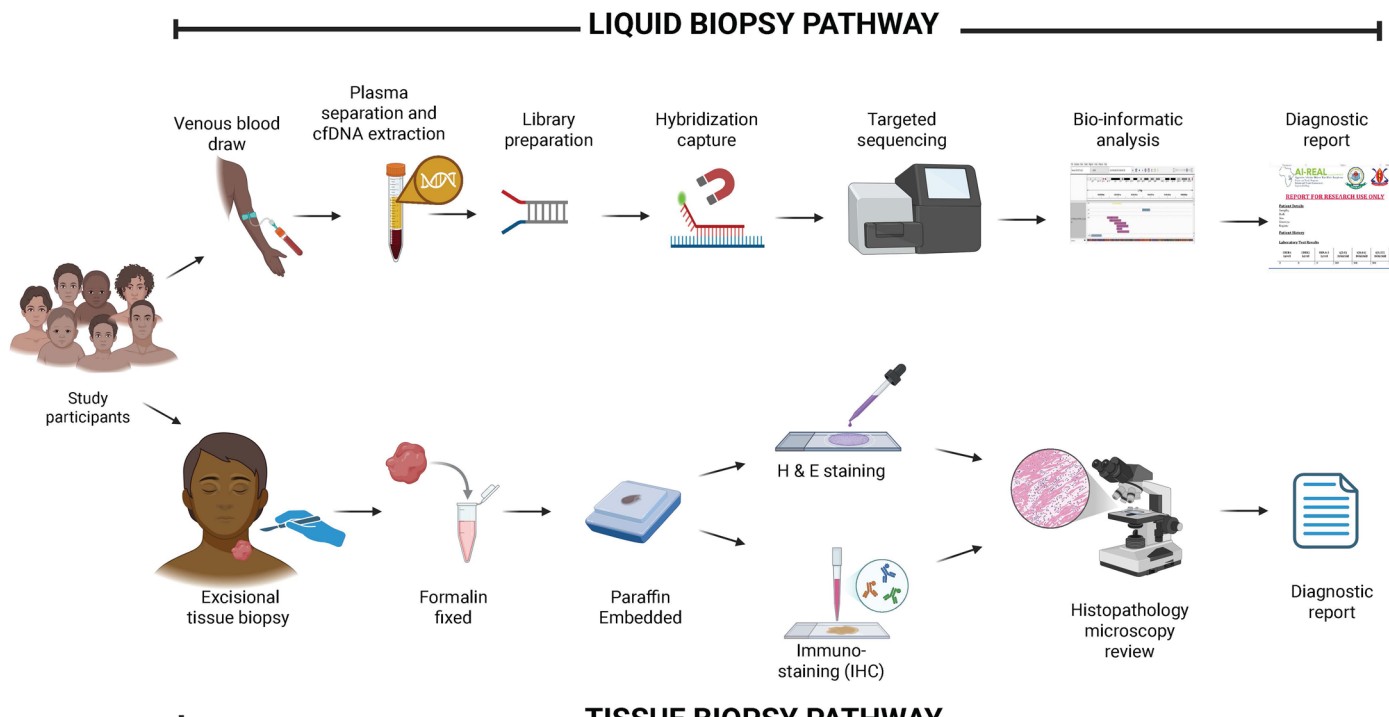

**Extended Data Fig. 7 | Liquid biopsy and tissue biopsy diagnostic workflows.** Schematic overview of parallel diagnostic pathways. The liquid biopsy pathway comprises venous blood collection, plasma separation and cfDNA extraction, library preparation, hybridization capture, targeted sequencing, bioinformatic analysis and generation of a diagnostic report. The tissue biopsy pathway comprises excisional biopsy, formalin fixation and paraffin embedding, haematoxylin and eosin staining, immunohistochemistry, histopathology review and diagnostic reporting.

**Extended Data Table 1 | Baseline demographic and clinical presentation of children and young adults with clinically suspected lymphoma (phase I and phase II),** *n*=377

| Characteristic | | Phase 1[1] (n=313) | Phase 2[1] (n=64) |
|---|---|---|---|
| **Age** | | 13 (9, 17) | 11 (7, 15) |
| **Sex** | Female | 106 (34%) | 19 (30%) |
| | Male | 207 (66%) | 45 (70%) |
| **Country of Origin** | Burundi | 1 (0.3%) | 0 (0%) |
| | Kenya | 1 (0.3%) | 0 (0%) |
| | South Sudan | 3 (1.0%) | 0 (0%) |
| | Tanzania | 222 (71%) | 39 (61%) |
| | Uganda | 86 (27%) | 25 (39%) |
| **Medical History** | HIV on ARV therapy | 11 (3.5%) | 3 (4.7%) |
| | Tuberculosis | 9 (2.9%) | 4 (6.3%) |
| **Family History of Cancer** | | 15 (4.8%) | 9 (14%) |
| **B Symptoms** | Weight loss | 218 (70%) | 50 (78%) |
| | Night sweats | 186 (61%) | 43 (68%) |
| | Fever | 209 (67%) | 39 (61%) |
| **Jaw mass** | | 78 (25%) | 8 (13%) |
| **Peripheral Lymph node enlargement** | | 213 (68%) | 34 (53%) |
| **Abdominal organomegaly** | Hepatosplenomegaly | 42 (13%) | 1 (1.6%) |
| | Hepatomegaly | 5 (1.6%) | 0 (0%) |
| | Splenomegaly | 18 (5.8%) | 8 (12.5%) |
| | Enlarged Kidney | 6 (1.9%) | 0 (0%) |
| | Unspecified abdominal mass | 96 (31%) | 16 (25%) |
| | No abdominal organomegaly | 146 (46.6%) | 39 (60.9%) |

[1]Median (IQR); n (%), HIV – Human Immunodeficiency Virus, ARV – Anti-Retroviral

Baseline demographic and clinical presentation of children and young adults with clinically suspected lymphoma (phase I and phase II), *n*=377.

**Extended Data Table 2 | Laboratory parameters and final diagnoses of children and young adults with clinically suspected lymphoma (phase I and phase II),** *n*=377

| Characteristic | Phase 1[1] (n=313) | Phase 2[1] (n=64) |
|---|---|---|
| **Laboratory Parameters** | | |
| Hemoglobin (g/dl) | 9.81 ( 8.15, 11.30) | 9.25 (7.98, 10.60) |
| Total White Blood Cell Count (x10$^9$/L) | 7.4 (5.2, 10.5) | 7.7 (5.8, 11.2) |
| Absolute neutrophil Count (x10$^9$/L) | 3.7 (2.5, 6.2) | 4.8 (3.1, 6.5) |
| Absolute Lymphocyte Count (x10$^9$/L) | 2.20 (1.33, 3.22) | 2.38 (1.58, 3.56) |
| Platelets (x10$^9$/L) | 300 (206, 437) | 323 (239, 439) |
| Lactate Dehydrogenase (IU/L) | 679 (393, 1,233) | 526 (350, 1,312) |
| **Final Diagnosis based on GSP tissue Biopsy** | | |
| Benign | 29 (9.3%) | 10 (15.6%) |
| Burkitt Lymphoma | 77 (24.6%) | 15 (23.4%) |
| Classical Hodgkin | 80 (25.6%) | 15 (23.4%) |
| Diffuse Large B-Cell | 45 (14.4%) | 5 (7.8%) |
| Nodular Lymphocyte Predominant Hodgkin | 1 (0.3%) | 1 (1.6%) |
| Non-Diagnostic | 33 (10.5%) | 8 (12.5%) |
| Other Cancer | 30 (9.6%) | 5 (7.8%) |
| Other lymphoma | 18 (5.8%) | 5 (7.8%) |

[1]Median (IQR); n (%), GSP — Gold Standard Pathology

Laboratory parameters and final diagnoses of children and young adults with clinically suspected lymphoma (phase I and phase II), *n*=377.

# Reporting Summary

## Statistics

For all statistical analyses, confirm that the following items are present in the figure legend, table legend, main text, or Methods section.

| n/a | Confirmed | |
|---|---|---|
| ☐ | ☒ | The exact sample size (*n*) for each experimental group/condition, given as a discrete number and unit of measurement |
| ☐ | ☒ | A statement on whether measurements were taken from distinct samples or whether the same sample was measured repeatedly |
| ☐ | ☒ | The statistical test(s) used AND whether they are one- or two-sided *Only common tests should be described solely by name; describe more complex techniques in the Methods section.* |
| ☐ | ☒ | A description of all covariates tested |
| ☐ | ☒ | A description of any assumptions or corrections, such as tests of normality and adjustment for multiple comparisons |
| ☐ | ☒ | A full description of the statistical parameters including central tendency (e.g. means) or other basic estimates (e.g. regression coefficient) AND variation (e.g. standard deviation) or associated estimates of uncertainty (e.g. confidence intervals) |
| ☐ | ☒ | For null hypothesis testing, the test statistic (e.g. *F*, *t*, *r*) with confidence intervals, effect sizes, degrees of freedom and *P* value noted *Give P values as exact values whenever suitable.* |
| ☒ | ☐ | For Bayesian analysis, information on the choice of priors and Markov chain Monte Carlo settings |
| ☒ | ☐ | For hierarchical and complex designs, identification of the appropriate level for tests and full reporting of outcomes |
| ☐ | ☒ | Estimates of effect sizes (e.g. Cohen's *d*, Pearson's *r*), indicating how they were calculated |

*Our web collection on statistics for biologists contains articles on many of the points above.*

## Software and code

Policy information about availability of computer code

| Data collection | Clinical and laboratory data were collected using electronic data capture with REDCap (version 13.1.0, © 2025 Vanderbilt University), hosted at Muhimbili National Hospital. Data were entered manually at the point of care using a tablet device. No commercial or custom code was used for data collection.  cfDNA sequence data was collected in FASTQ format and stored by a bespoke pipeline in a HIPPAA compliant, AWS cloud. |
|---|---|
| Data analysis | Clinical and laboratory data analysis was performed in R version 4.2.3 using open-source packages.  Sequence data was analsyed in a bespoke pipeline. Analysis incorporated both custom tools (udini, elduderino, trim) and widely used open-source bioinformatics software (Samtools, BWA-MEM2, Varscan, VEP, IgCaller, GRIDSS)  All codes used in the analysis are accessible at https://github.com/ClaraClaudius/CLINICAL-VALIDATION-OF-LIQUID-BIOPSY-FOR-FASTER-DIAGNOSIS-OF-EBV-POSITIVE-BURKITT-LYMPHOMA.git. Tools used for the bioinformatics analysis are as follows: <br> 1. Udini - This is a locally created tool. It extracts UMIs from FASTQ files. It combines all of the reads for a sample into one interleaved FASTQ. It also puts QC information into the stats.json file, specifically,total reads and reads that are invalid for being too short (threshold is 50bp) <br> 2. BWA-MEM2 (version 2.2.1) - This is an open-source reimplementation of BWA-MEM, was used for aligning sequencing reads to the GRCh37 reference genome <br> 3. Elduderino - This is a locally created tool.  It is used to de-duplicate the read. Read families are collapsed into consensus reads. Elduderino also adds smetrics to the stats.json file, including, error rates, and family sizes <br> 4. size - Locally developed script. Uses the read pairs to calculate fragment sizes and saves fragment sizes to stats.json and plotted in the sizes.pdf file <br> 5. ontarget - locally created tool. This takes the deduplicated, and re-aligned, SAM file along with a bed file as it's input; If any of the read overlaps with any region of interest in the bed file then that read is kept. This is also were levels of EBER1, EBER2 and EBNA-2 are calculated. The mean depth of coverage for each EBV gene is divided by the mean depth for all other genes to give an approximation of the number of |

copies present relative to other genes. Also adds off target rate to stats.json file

6. trim - Trims the last base from the end of each read improving the quality of variant calling

7. covermi_stats - This is a locally developed python script, which uses the locally developed CoverMi package. It takes BAM and panel folder as input. The panel folder contains the bed file as well as some information about which transcripts to use and exon locations. Generates coverage statistic and calculates the mean depth across all targets as well as the percentage of targets that were covered at 30, 100, 500, 1000 and 2000x. It also gives this statistics on a per gene basis

8. call_variants - locally developed python script.This is essentially a wrapper script that is used to easily call multiple different variant callers as well as annotating the resulting VCF files with ensembls variant effect predictor (VEP). Has many inputs including the BAM file, reference genome, variant caller to use (current options are vardict, varscan and mutect2) along with the minimum VAF and minimum number of alternate reads required to call a variant

9. vcf_stats - custom python script - This provides some statistics about the VCFs including total number of variants, total number of insertions and deletions, or InDels, the transition to transversion or ti/tv ratio. All of this metrics get added to the stats.json file

10. IgCaller (Version 1.2 software utilizing the hg19 reference genome)- Python script designed to fully characterize the immunoglobulin gene rearrangements and oncogenic translocations in lymphoid neoplasms.

11. GRIDSS (the Genomic Rearrangement IDentification Software Suite) version 2.13.2 - structural variant caller

For manuscripts utilizing custom algorithms or software that are central to the research but not yet described in published literature, software must be made available to editors and reviewers. We strongly encourage code deposition in a community repository (e.g. GitHub). See the Nature Portfolio guidelines for submitting code & software for further information.

# Data

Policy information about availability of data

All manuscripts must include a data availability statement. This statement should provide the following information, where applicable:
- Accession codes, unique identifiers, or web links for publicly available datasets
- A description of any restrictions on data availability
- For clinical datasets or third party data, please ensure that the statement adheres to our policy

The raw sequencing data and individual-level clinical data generated in this study cannot be made publicly available owing to ethical and data protection restrictions, as they include information from human participants collected under institutional approvals that do not permit open public deposition. Data access is governed by the study consortium and collaborating centers in Tanzania and Uganda. Requests to access the underlying anonymized data will be reviewed by the consortium's data access committee in consultation with all participating institutions.

If a request is deemed scientifically sound and compliant with applicable institutional, national, and international data protection regulations, de-identified and anonymized data will be shared following the execution of a data transfer agreement. Processed and anonymized data supporting the findings of this study are available in the GitHub repository  https://github.com/ClaraClaudius/CLINICAL-VALIDATION-OF-LIQUID-BIOPSY-FOR-FASTER-DIAGNOSIS-OF-EBV-POSITIVE-BURKITT-LYMPHOMA.git. This repository constitutes the source data for this paper and is provided to ensure transparency and reproducibility. All data-sharing requests should be addressed to the corresponding author (clarachamba@gmail.com). Timelines for review, approval, and data transfer may vary depending on the scope and regulatory requirements of each request

# Research involving human participants, their data, or biological material

Policy information about studies with human participants or human data. See also policy information about sex, gender (identity/presentation), and sexual orientation and race, ethnicity and racism.

| Reporting on sex and gender | Sex (a biological attribute) was recorded for all participants based on clinical records. Gender identity was not assessed. The study included both male and female participants. Sex was evaluated in a univariate analysis using the chi-squared test to assess its association with Burkitt lymphoma diagnosis, and was also included as a covariate in the multivariable diagnostic models. The results of these analyses are reported in the main text and source data. Consent for sharing de-identified sex-disaggregated data was not obtained; therefore, only aggregate results are reported |
|---|---|
| Reporting on race, ethnicity, or other socially relevant groupings | The only socially relevant categorization variable used in this study was sex, recorded as male or female based on clinical records. Gender identity, race, ethnicity, and socioeconomic status were not collected or analyzed. Sex was included as a variable to assess potential biological associations with diagnosis and was incorporated into both univariate and multivariable models. No proxy variables were used in place of other social constructs. Confounding was addressed by including clinically relevant covariates in multivariable logistic regression models, including age, LDH, EBV status, MYC-related molecular features, and cfDNA levels |
| Population characteristics | Covariate-relevant characteristics of the study participants included clinical and molecular variables collected to assess their diagnostic relevance and association with Burkitt lymphoma (BL). Clinical variables included age, sex, LDH level, duration of symptoms, and presence of jaw or abdominal mass. Molecular variables incorporated into the analysis included: median variant allele frequency (VAF) of somatic mutations in MYC, TP53, and ID3; circulating tumor DNA (ctDNA), calculated as the product of cfDNA concentration and median VAF; the absolute number of MYC intron 1 and exon 2 mutations; and EBV-related features including EBER1, EBER2, and EBNA2 copy number (copies per cell), the maximum value of EBV copies per cell (EBVmax), EBV fragment size ratio (EBVSR), EBV DNA proportion (EBVP), and entropy measures for EBV and autosomal DNA fragments. These variables were derived from targeted next-generation sequencing data and used as covariates in univariate and multivariable models to identify predictors of BL diagnosis |
| Recruitment | Participants were recruited prospectively from four referral hospitals in Tanzania and Uganda (Muhimbili National Hospital, Kilimanjaro Christian Medical Centre, Bugando Medical Centre, and St. Mary's Hospital Lacor) between August 2019 and July |

2023. Inclusion was limited to children and young adults aged 3–25 years with a clinical suspicion of lymphoma who provided informed consent. Patients who had previously received chemotherapy, immunotherapy, investigational agents, or participated in clinical trials were excluded. As recruitment was limited to referred cases presenting to tertiary centers, there is a potential for referral bias, which may overrepresent patients with more advanced or complex disease presentations. Self-selection bias is unlikely, as patients were enrolled consecutively based on clinical eligibility and willingness to consent. However, the exclusion of pretreated cases may limit generalizability to treatment-naive populations

| | |
|---|---|
| Ethics oversight | This study was approved by the Oxford Tropical Research Ethics Committee (OxTREC: 15-19), the National Institute of Medical Research (NIMR) in Tanzania (NIMR/HQ/R.8a/Vol.IX/3408), and the Uganda National Council of Science and Technology (UNCST: HS529ES) |

Note that full information on the approval of the study protocol must also be provided in the manuscript.

# Field-specific reporting

Please select the one below that is the best fit for your research. If you are not sure, read the appropriate sections before making your selection.

☒ Life sciences ☐ Behavioural & social sciences ☐ Ecological, evolutionary & environmental sciences

For a reference copy of the document with all sections, see nature.com/documents/nr-reporting-summary-flat.pdf

# Life sciences study design

All studies must disclose on these points even when the disclosure is negative.

| | |
|---|---|
| Sample size | A formal sample size calculation was conducted to ensure sufficient power for evaluating diagnostic performance. Using the binomial proportion formula with Z=1.96 (95% confidence), p=0.8 (expected sensitivity), and d=0.10 (margin of error), the minimum number of positive cases required was estimated to be 62, corresponding to a total sample size of approximately 124 participants (assuming a 1:1 case-to-control ratio). Additionally, using the events-per-variable (EPV) method with a conservative threshold of 20 and up to 18 predictors, we estimated a total required sample size of 720 participants. However, due to the prospective nature of the study and unforeseen disruptions during the COVID-19 pandemic—including delays in clinical workflows and research staffing—we were unable to reach this target. The final cohort included 212 participants, of whom 81 were diagnosed with Burkitt lymphoma. To address this limitation, we employed LASSO (Least Absolute Shrinkage and Selection Operator) regression, which performs regularization and variable selection, thereby reducing over fitting in models with relatively small sample sizes and many candidate predictors. We also used 10-fold cross-validation to internally validate model performance and assess generalizability. These strategies align with best practices for model development and transparent reporting in constrained-sample settings, as recommended by the TRIPOD guidelines. |
| Data exclusions | Participants with missing outcome data were excluded from the analysis. This exclusion was predefined to maintain the integrity of outcome-based comparisons. Only individuals lacking key clinical endpoints or follow-up information necessary for primary analysis were removed prior to statistical evaluation. For other missing data, such as covariates, appropriate imputation methods were applied to minimize bias and retain statistical power. |
| Replication | We took several measures to ensure the reproducibility of our findings. All sequencing and data processing were performed using standardized protocols and quality control criteria applied uniformly across samples. Bioinformatic analyses were conducted using version-controlled pipelines and documented custom scripts, with core steps (e.g., alignment, variant calling, EBV quantification) implemented through reproducible workflows.For model development and evaluation, we used 10-fold cross-validation to verify internal reproducibility and reduce overfitting. All statistical analyses were scripted in R and Python, and the full codebase is publicly available along with the source data used to generate the results.All attempts to replicate the main findings across cross-validation folds and within subgroups were successful and consistent. There were no results in this study that could not be reproduced. |
| Randomization | This was a prospective observational study. Participants were not randomly allocated into experimental groups. Instead, diagnostic groupings (e.g., Burkitt lymphoma vs. non-Burkitt lymphoma) were based on final clinical and pathological diagnoses. As such, group assignment reflected natural clinical presentation rather than study-driven allocation. To control for potential confounding, relevant clinical and molecular covariates (e.g., age, LDH, EBV status) were included in multivariable regression models and penalized regression techniques (LASSO) were applied during model development. |
| Blinding | Blinding of investigators was not applicable during participant recruitment or diagnostic group assignment, as group status was determined retrospectively based on clinical and pathological findings. However, data analysis—including cfDNA quantification, variant calling, and model development—was performed using automated pipelines and pre-specified statistical workflows, reducing the potential for bias. Investigators conducting statistical analysis were aware of group labels to enable supervised model training. Given the objective nature of the molecular measurements and algorithm-driven methods used, the absence of blinding was not expected to influence study outcomes. |

# Reporting for specific materials, systems and methods

We require information from authors about some types of materials, experimental systems and methods used in many studies. Here, indicate whether each material, system or method listed is relevant to your study. If you are not sure if a list item applies to your research, read the appropriate section before selecting a response.

## Materials & experimental systems

| n/a | Involved in the study |
|---|---|
| ☒ ☐ | Antibodies |
| ☒ ☐ | Eukaryotic cell lines |
| ☒ ☐ | Palaeontology and archaeology |
| ☒ ☐ | Animals and other organisms |
| ☒ ☐ | Clinical data |
| ☒ ☐ | Dual use research of concern |
| ☒ ☐ | Plants |

## Methods

| n/a | Involved in the study |
|---|---|
| ☒ ☐ | ChIP-seq |
| ☒ ☐ | Flow cytometry |
| ☒ ☐ | MRI-based neuroimaging |

## Plants

| | |
|---|---|
| Seed stocks | *Report on the source of all seed stocks or other plant material used. If applicable, state the seed stock centre and catalogue number. If plant specimens were collected from the field, describe the collection location, date and sampling procedures.* |
| Novel plant genotypes | *Describe the methods by which all novel plant genotypes were produced. This includes those generated by transgenic approaches, gene editing, chemical/radiation-based mutagenesis and hybridization. For transgenic lines, describe the transformation method, the number of independent lines analyzed and the generation upon which experiments were performed. For gene-edited lines, describe the editor used, the endogenous sequence targeted for editing, the targeting guide RNA sequence (if applicable) and how the editor was applied.* |
| Authentication | *Describe any authentication procedures for each seed stock used or novel genotype generated. Describe any experiments used to assess the effect of a mutation and, where applicable, how potential secondary effects (e.g. second site T-DNA insertions, mosiacism, off-target gene editing) were examined.* |

