## [Peer Review File · Nature Medicine]

LIQUID BIOPSY FOR THE DIAGNOSIS OF EBV-POSITIVE BURKITT LYMPHOMA IN ENDEMIC AREAS

Corresponding Author: Dr Clara Chamba

Version 1:

Reviewer comments:

Reviewer #1

(Remarks to the Author)

This is a very interesting and original work on the use of ctDNA in the strategy of initial diagnosis of Burkitt lymphoma in endemic countries.

This strategy is based on the almost constant presence of EBV in those Burkitt lymphoma in the endemic form and classical genetic molecular abnormalities in this lymphoma (Myc, ID3 and P53). The combination of Burkitt genetic alterations and EBV presence seems to give a robust diagnostic tool in few days and compare very favorably with the standard immunohistochemistry analysis in pathology which take weeks. A prospective cohort validate the initial data generated in the first cohort of more that 200 patients.

It could be interesting to know what is the panel of antibodies used in the IHC "gold standard" method for diagnosis.

It could be also interesting to have some clinical characteristics of the patients who had a Burkitt lymphoma on IHC but no ctDNA abnormalities in favor of the diagnosis.

Reviewer #2

(Remarks to the Author)

The Chamba et al. manuscript presents clinical validation data of a novel, liquid biopsy based approach for fast(er) molecular diagnosis of EBV+ Burkitt's lymphoma. Although technically the approach is not novel, I like several aspects: the fact that the developed models are explainable, that they are based on features that are amenable to acquire in the endemic region and that the circuit has been conceived and developed with very clear clinical priorities in mind. It is true that a small sequencing instrument is required as well as molecular biology reagents and trained technical personnel, but this is largely easier than having trained pathologists. And is also quite faster and objective in evaluation.

I have a couple of doubts/comments,

-suppl. figure 2: mutations appear to be clustered in some positions and others are void of events, how is the typical profile of coverage in MYC's intron? in addition, MYC-IG translocation points could also be plotted and shown?

-Fig.3: could authors add on the figure the percentage of occasions in which MDTs had to decide based on each of the 5 scenarios for BL diagnosis in the 'real-life' stage II setting?

-Table 3: some features in table 3 could be converted into a figure format, making it more amenable for the readers (ex. LB values, EBV related values).

-line 356: could authors disclose and later on discuss how many missing values needed be imputed in Stage II? this is a limitation to the approach and is important to address and evaluate

-Fig. 6/6 TB cases resulted in a non-BL diagnosis. these are small numbers of cases, but is there any possible explanation for this bias?

Reviewer #3

(Remarks to the Author)

This manuscript describes the validation of a liquid biopsy and associated Comprehensive Model for the rapid diagnosis of

Burkitt Lymphoma (BL) in East Africa. BL is a common childhood malignancy in Africa. It is an extremely aggressive malignancy and rapid diagnosis is essential to guide management. The manuscript sets out the existing problems with applying standard histological approaches and the need for quicker and more reliable diagnostic approaches. The authors' liquid biopsy assay was incorporated into a Comprehensive Model developed in a discovery cohort and then validated in a prospective validation cohort. This approach showed impressive sensitivity/specificity rates of 83%/95% respectively. This suggests that this approach could overcome existing diagnostic challenges for BL in developing countries. The turnaround times and diagnostic accuracy are extremely impressive. The incorporation into diagnostic MDTs, the large cohort size and the very considerable overall impact of time to diagnosis for some patients are notable strengths of this study. It is important to see the benefit of NGS extended to developing countries where, as this manuscript shows, it has potential to make much more tangible and immediate impacts that it might make in a first world setting, in which the vast majority of studies are performed.

However, I have a few concerns.

The targeted sequencing assay was developed and described in a recent publication (JCO Global Oncol Jan 2025). The authors now validate its performance in a larger, prospective cohort. However, it is not clear that the assay is fully optimized. The pathognomonic genetic feature of BL is rearrangement of MYC. This is only detected in 48% of BL cases in the current study. Why is the detection rate so low and did the authors investigate the features of cases with failed detection? Could this be improved by additions to the panel or deeper sequencing? Other studies have suggested much higher pickup of MYC rearrangement from ctDNA, at least in DLBCL (eg Scherer, Sci Trans Med 2016).

The calculation of average VAF from a small number of variants could be problematic since the genes targeted are often subject to copy number change – a loss of one allele of TP53 could artificially inflate the VAF of a mutation on the other allele. Similarly, MYC amplification is often seen in non-BL NHL. This might be important since the average VAF between BL and non-BL seems only 2-fold different in Table 3 (0.04 vs 0.02). How have the authors accounted for the impact of copy number changes on the calculation of VAF. The authors provide overall summary metrics for average VAF, but given that this is a critical metric to the model, they should provide individual average VAF for each case, grouped by diagnosis, along with the VAFs of individual SNVs. This would be important to show readers how often the average VAF is based upon a single variant, and how often a VAF is called in non-malignant cases.

Somatic hypermutation of intron 1 may be one way to infer the presence of a MYC translocation. However, the mutation counts are surprisingly high – mean 54 mutations/case in BL and 26/case in non-BL. I am surprised that there are any mutations at all in non-BL without rearrangement of MYC. Are these truly somatic mutations? Could the panel be detecting single nucleotide polymorphisms. These will be more prevalent in non-coding (intronic) regions. I do not think matched germline DNA was used to exclude SNPs and the discussion raises the important issue of the poor documentation of the reference genome in African countries. If matched GL DNA is not available, have the authors shown absence of these mutations in a panel of normal plasmas? It would only need one or two SNPs to escape filtering to make a big impact on average VAF.

Given the above difficulty to exclude SNPs in intronic regions, and the frequent missing of IgH rearrangements, how would the model handle a case of acute EBV infection with high EBV titer, splenomegaly and cervical lymphadenopathy? Might this pick up enough factors on the Comprehensive model to score as BL? Since the MYC rearrangement is only found in 48% cases there is no requirement to find this. There are no other exonic variants in Fig4c and the VAF could be falsely contributed by intronic SNPs. Would the EBV DNA / latency profile in acute EBV infection be sufficiently different from BL to avoid scoring into the comprehensive model? Has the model been tested, and validated as negative, in cases of acute EBV infection without BL?

I am unclear what was included in the gold standard pathology analysis. The authors cite a BJHaem paper, but I can't see what IHC stains were used. It also cites an 81% diagnostic accuracy of that IHC panel. Was the same panel used for GSP in this study? If so, if the gold standard is only 81% accurate, is it meaningful to distinguish between the 5 models tested?

The workings of the models are a black box. The supplementary figure is helpful in understanding the relative contribution of each element of each model, but it's still not completely clear how different factors influence the output. It does seem that the dominant factors to the overall model are EBV proportion. There might be easier, quicker, cheaper ways to measure this – like qPCR. Since this seems to be the main factor contributed by the liquid biopsy, why is a targeted sequencing panel needed at all? Would a similar model using EBV DNA qPCR plus clinical factors be as good? The only two factors that appear to be contributed by the targeted sequencing are the average VAF (see comment 2) and MYC translocation (which is missed in 52% of cases).

The turnaround times in this study are vastly better for liquid biopsy than for IHC. It is not clear to me why this should be the case. In my experience it is surprisingly difficult to achieve this kind of TAT for NGS in a first world context, requiring technical expertise. Why is this easier in Africa? Conversely, why is the IHC TAT so poor? Shown as 6 months in some cases. This is not a criticism of the experimental design - I'm just trying to understand better. Is it that it is easier to train staff to run NGS assays than it is to perform and interpret IHC?

Version 2:

Reviewer comments:

Reviewer #1

(Remarks to the Author)

the manuscript is now really improved by the modifications done by the authors after review. Important data are presented.

Reviewer #2

(Remarks to the Author)

Authors have provided appropriate responses to my questions/comments.

Reviewer #3

(Remarks to the Author)

Thanks to the authors for responding so comprehensively to my many comments and for the extra analysis that has been performed. I am satisfied that all my comments have been addressed and that limitations have been appropriately discussed.

The manuscript makes a very important point - that liquid biopsy improves the diagnostic turnaround time for endemic Burkitt lymphoma in sub-Saharan Africa. Given the perceived technical difficulties of liquid biopsy and NGS, it was not obvious that this would be the case. In this sense, the results are important, novel and really remarkable. The findings will be of interest to a broad range of clinicians and scientists. I fully support publication of the manuscript in its current form.

Confidential Information Redacted

Referee #1: Clinical Burkitt lymphoma

Referee #2: Liquid biopsies

Referee #3: Lymphomas, clinical and genetics

Reviewers' Comments:

Reviewer #1 (Remarks to the Author):

This is a very interesting and original work on the use of ctDNA in the strategy of initial diagnosis of Burkitt lymphoma in endemic countries.

This strategy is based on the almost constant presence of EBV in those Burkitt lymphoma in the endemic form and classical genetic molecular abnormalities in this lymphoma (Myc, ID3 and P53). The combination of Burkitt genetic alterations and EBV presence seems to give a robust diagnostic tool in few days and compare very favorably with the standard immunohistochemistry analysis in pathology which take weeks. A prospective cohort validate the initial data generated in the first cohort of more than 200 patients.

It could be interesting to know what is the panel of antibodies used in the IHC "gold standard" method for diagnosis.

RESPONSE: Thank you for this valuable comment. We have clarified this point by providing a detailed description of the immunohistochemistry (IHC) antibody panel used in the gold standard diagnostic method. This information has been added to the Materials and Methods section under the sub-section Local Gold Standard Pathology (page 6). The revised sentence reads as follows:

This diagnostic approach utilizes a three-phase scoring system. The first phase combines typical Burkitt lymphoma (BL) morphology with immunostains BCL2, CD10, and CD20 to establish a diagnosis. Cases that remain inconclusive proceed to a second phase with CD38, CD44, and Ki67, while unresolved cases undergo a third phase with FISH analysis for MYC-Ig translocation. In our study, we used the first-phase immunostains (BCL2, CD10, and CD20) to support the diagnosis of BL, as this phase has previously been shown to achieve 81% concordance with WHO SoC pathology and is well-suited for the limited-resource settings (Naresh KN et al. British journal of haematology 2011)

It could be also interesting to have some clinical characteristics of the patients who had a Burkitt lymphoma on IHC but no ctDNA abnormalities in favor of the diagnosis.

RESPONSE: Thank you for this insightful comment. Among 81 patients diagnosed with Burkitt lymphoma by Local Gold Standard Pathology, 11 (13.6%) were not detected by the liquid biopsy model (false negatives), while 70 (86.4%) were correctly classified (true positives). The clinical characteristics of the false-negative cases (those with Burkitt lymphoma confirmed by IHC but without ctDNA abnormalities) are now summarized in Supplementary Table 4. These results have also been incorporated into the Results section (page 11) and reads as follows:

Among 81 patients with a confirmed local gold standard diagnosis of BL, 11 (13.6%) were not detected by the liquid biopsy model (false negatives), while 70 (86.4%) were correctly identified (true positives). False-negative cases were comparable to true positives with respect to age, sex distribution and LDH levels. However, they were significantly less likely to present with jaw or abdominal tumours ($p < 0.05$) (Supplementary Table 4).

Reviewer #2 (Remarks to the Author):

The Chamba et al. manuscript presents clinical validation data of a novel, liquid biopsy based approach for fast(er) molecular diagnosis of EBV+ Burkitt's lymphoma. Although technically the approach is not novel, I like several aspects: the fact that the developed models are explainable, that they are based on features that are amenable to acquire in the endemic region and that the circuit has been conceived and developed with very clear clinical priorities in mind. It is true that a small sequencing instrument is required as well as molecular biology reagents and trained technical personnel, but this is largely easier than having trained pathologists. And is also quite faster and objective in evaluation. I have a couple of doubts/comments,

-suppl. figure 2: mutations appear to be clustered in some positions and others are void of events, how is the typical profile of coverage in MYC's intron? in addition, MYC-IG translocation points could also be plotted and shown?

RESPONSE: Our panel covered the MYC 5' region as well as MYC exon 1, intron 1 and exon 2 (supplementary Table 1, row 5- 8). A representative BL sample is shown in Supplementary Figure 2, illustrating the regions covered by the probes (black bars) and a typical coverage profile. Across the cohort, the median, mean depth of coverage was 1586X (IQR: 1,156 - 2,234) (supplementary table 5).

We have also repeated the mutation distribution plot for BL and non-BL cases with exclusion of variants in homopolymer regions (supplementary Fig 6A and B). Note that this figure has changed as we have revised the variant filtering criteria for calling of somatic mutations following comments raised by reviewer no.3 (page 7 of the revised manuscript - track changes version). This clearly shows the different distribution of mutations in MYC in BL and non-BL cases.

Breakpoint Locations on the MYC and IGH loci have been plotted showing percentage of samples with breakpoints in different locations. This has been added as supplementary Fig. 5 and quoted in the revised manuscript file on page 11 reading as follows:

The majority of MYC-IGH breakpoints on the MYC locus occurred within the class II (46%, 18/39) and class III (28%, 11/39) regions. On the IGH locus, 51% (20/39) of breakpoints were located within the VDJ junction region (Supplementary Fig. 5).

-Fig.3: could authors add on the figure the percentage of occasions in which MDTs had to decide based on each of the 5 scenarios for BL diagnosis in the 'real-life' stage II setting?

RESPONSE: Thank you for this helpful suggestion. We have now added the percentages representing the frequency of decisions made under each of the five diagnostic scenarios in Figure 3 (page 9). The figure legend has also been revised accordingly to reflect these additions.

-Table 3: some features in table 3 could be converted into a figure format, making it more amenable for the readers (ex. LB values, EBV related values).

RESPONSE: We have added a supplementary figure for all the EBV markers showing their distribution stratified by diagnosis. This is now supplementary figure 4.

-line 356: could authors disclose and later on discuss how many missing values needed be imputed in Stage II? this is a limitation to the approach and is important to address and evaluate

RESPONSE:

We appreciate this comment. The number of missing values and the imputation method were already described in the Methods section under Data Analysis on page 8. Specifically, we stated: “In this analysis, missing data for Lactate Dehydrogenase (LDH) in 12 samples was imputed using multiple imputation by Chained Equations (MICE), implemented in the mice R package (v3.16.0). Predictive mean matching (method = ‘pmm’) was applied.” Figure 5 also presents both results including imputed LDH values (Fig. 5A) and excluding them (Fig. 5B), allowing assessment of any impact of imputation on model performance.

-Fig. 6/6 TB cases resulted in a non-BL diagnosis. these are small numbers of cases, but is there any possible explanation for this bias?

RESPONSE Thank you for this observation. In the Sankey plot (Fig. 6), the trajectories of all six TB cases are shown—one case ultimately flowed towards a final BL diagnosis, while the remaining five (5/6) were classified as non-BL. This is visually represented by the connecting strands in the diagram linking TB to BL and non-BL nodes. We have clarified this point in the text under the subsection Integration of the Liquid Biopsy Test into the Diagnostic Pathway through a Multi-Disciplinary Meeting (page 13), which now states: “Among all cases that were finally diagnosed as BL (n = 15), in the first MDT, 1 (6.7%) was diagnosed using tissue biopsy alone...”

Reviewer #3 (Remarks to the Author):

This manuscript describes the validation of a liquid biopsy and associated Comprehensive Model for the rapid diagnosis of Burkitt Lymphoma (BL) in East Africa. BL is a common childhood malignancy in Africa. It is an extremely aggressive malignancy and rapid diagnosis is essential to guide management. The manuscript sets out the existing problems with applying standard histological approaches and the need for quicker and more reliable diagnostic approaches. The authors’ liquid biopsy assay was incorporated into a Comprehensive Model developed in a discovery cohort and then validated in a prospective validation cohort. This approach showed impressive sensitivity/specificity rates of 83%/95% respectively. This suggests that this approach could overcome existing diagnostic challenges for BL in developing countries. The turnaround times and diagnostic accuracy are extremely impressive. The incorporation into diagnostic MDTs, the large cohort size and the very considerable overall impact of time to diagnosis for some patients are notable strengths of this study. It is important to see the benefit of NGS extended to developing countries where, as this manuscript shows, it has potential to make much more tangible and immediate impacts that it might make in in first world setting, in which the vast majority of studies are performed.

However, I have a few concerns.

The targeted sequencing assay was developed and described in a recent publication (JCO Global Oncol Jan 2025). The authors now validate its performance in a larger, prospective cohort. However, it is not clear that the assay is fully optimized. The pathognomonic genetic feature of BL is rearrangement of MYC. This is only detected in 48% of BL cases in the current study. Why is the detection rate so low and did the authors investigate the features of cases with failed detection? Could this be improved by additions to the panel or deeper sequencing? Other studies have suggested much higher pickup of MYC rearrangement from ctDNA, at least in DLBCL (eg Scherer, Sci Trans Med 2016).

RESPONSE: Thank you for this insightful comment. Our targeted assay was intentionally designed to be cost-effective. At the MYC locus we targeted exon 2, intron 1, exon 1 and the region approximately 1.6kb upstream of exon 1. Together, based on previously published

data (schema below; Joos S et al), these regions are targeted by translocations in about 40% of cases of EBV-positive BL. In the other 60% of cases, MYC breakpoints are frequently dispersed tens to hundreds of kilobases 5' to MYC, whereas EBV-negative BL has a more proximal distribution (see image below from Lopez C et al). Consequently, a substantial fraction of EBV-positive cases are expected to fall outside our capture window, which we believe accounts for most of the non-detections. Covering the entire upstream MYC locus would have compromised cost-effectiveness. With our panel design, we therefore expect to capture about 40% of the breakpoints, while fully covering established mutation hotspots in MYC that act as a surrogate for translocations as we and others previously demonstrated (Chamba C et al *Global Oncology* 2025; Cucco F et al *Leukemia* 2020).

The detection of MYC-IGH translocations is typically higher in DLBCL, where breakpoints are confined to a narrower genomic region compared with the more dispersed breakpoint distribution characteristic of BL (Chong et al., *Blood Adv.*, 2018).

Upon comparison of sequencing metrics between translocation-positive and translocation-negative cases, we observed that translocation-positive cases demonstrated slightly higher mean sequencing depth overall, as well as for the MYC gene and 5' MYC region, although the median depth for both groups exceeded 1000× (Supplementary Table 5). This difference in depth is therefore unlikely to account for the absence of detectable translocations in some cases, as coverage in both groups was well above the threshold required for reliable detection.

While extending the panel design to cover several hundred kilobases upstream of MYC could potentially improve detection sensitivity in EBV⁺ BL, this would substantially increase sequencing costs and data burden, counteracting the goal of developing a low-cost, targeted assay suitable for routine clinical use. This clarification has been added to the Discussion (page 16-17), reading as follows:

Due to the rational panel design, the MYC-IGH translocation was detectable in only 48% of BL cases. Although translocation-positive cases showed slightly higher sequencing depth, the median coverage across both groups exceeded 1000× (supplementary table 5), indicating that insufficient depth is unlikely to explain the absence of detectable MYC translocations in some BL cases. Instead, this likely

reflects the broad and heterogeneous distribution of MYC breakpoints in BL, which frequently extend beyond the genomic region captured by our current panel. Expanding the targeted region could enhance detection sensitivity but would increase sequencing costs and data complexity, contrary to the assay's objective of remaining a cost-efficient and scalable diagnostic tool for use in low-resource settings.

The calculation of average VAF from a small number of variants could be problematic since the genes targeted are often subject to copy number change – a loss of one allele of TP53 could artificially inflate the VAF of a mutation on the other allele. Similarly, MYC amplification is often seen in non-BL NHL. This might be important since the average VAF between BL and non-BL seems only 2-fold different in Table 3 (0.04 vs 0.02). How have the authors accounted for the impact of copy number changes on the calculation of VAF.

RESPONSE: We thank the reviewer for raising this important point. We did not directly test for copy number alterations (CNAs) in this study as this would have been cost prohibitive as it requires tiling across regions of interest. Instead, our targeted sequencing panel was primarily designed to detect BL driver mutations, MYC translocations and EBV. Moreover, the relatively short and uneven genomic coverage of targeted capture panels limits reliable CNA calling, particularly in circulating cell-free DNA where tumor-derived fragments represent a small and variable fraction of total cfDNA. Nonetheless, we recognized the potential confounding effect of CNAs on variant allele frequency (VAF) estimation. To mitigate this, we used an internal normalization approach. To account for the influence of copy number alterations on VAF, we first established an expected relationship between VAF and circulating tumour DNA (ctDNA). This was derived from ID3 mutations, which are recurrent in BL but rarely subject to copy number alterations. A simple linear regression of VAF on \log_{10} -transformed ctDNA was fitted for ID3, and predicted VAF values were used to calculate residuals (observed minus expected). A tolerance band was defined as ± 2 times the median absolute deviation of the ID3 residuals. This approach was then applied to both MYC and TP53 variants, with those falling outside the band flagged as likely CNA-affected (inflated by loss or diluted by amplification) (Supplementary Fig. 1A and 1B). Supplementary table 2 summarises these variants by gene, position and diagnosis. These variants have now been excluded in all downstream analysis. Thanks to the correction for CNAs, the difference in VAF between BL and non-BL samples has improved further and is now 4-fold (0.07 vs 0.02) and not 2-fold (0.04 vs 0.02) (Table 3) as before. These details have been added in the methods section on page 7 (track changes version) and read as follows:

Copy number alterations (CNA): We excluded variants that showed evidence of being affected by copy number alterations. This was done through an internal normalisation approach in which the expected relationship between VAF and \log_{10} -transformed ctDNA was modelled using ID3 mutations (rarely affected by CNAs), and the resulting tolerance limits ($\pm 2 \times$ median absolute deviation of residuals) were applied to MYC and TP53 variants to flag likely CNA-affected outliers (Supplementary Fig. 1A–B, supplementary table 2).

The authors provide overall summary metrics for average VAF, but given that this is a critical metric to the model, they should provide individual average VAF for each case, grouped by diagnosis, along with the VAFs of individual SNVs. This would be important to show readers how often the average VAF is based upon a single variant, and how often a VAF is called in non-malignant cases.

RESPONSE: We thank the reviewer for this helpful suggestion. We have added supplementary Fig. 1C, that shows the distribution of VAFs for all individual SNVs within each sample that had a mutation, grouped by diagnosis (BL vs non-BL). In this plot, each dot

represents an individual variant, while the dark blue dot denotes the median VAF for that specific sample. Group medians for BL and non-BL are indicated with solid lines, and the overall difference was tested using the Wilcoxon test ($p < 0.001$). Note that these values are different from the previously submitted manuscript as we have now excluded all variants that may be affected by copy number alterations in MYC and TP53 and also applied additional filtering criteria based on reviewers comment below (page 7). To address the reviewer's concern about how often a median VAF is based on a single variant, the x-axis is annotated with the number of variants contributing to each sample. This will allow readers to readily appreciate when the sample median reflects only one variant and when it is based on multiple SNVs, including in non-BL cases.

Somatic hypermutation of intron 1 may be one way to infer the presence of a MYC translocation. However, the mutation counts are surprisingly high – mean 54 mutations/case in BL and 26/case in non-BL. I am surprised that there are any mutations at all in non-BL without rearrangement of MYC. Are these truly somatic mutations? Could the panel be detecting single nucleotide polymorphisms. These will be more prevalent in non-coding (intronic) regions. I do not think matched germline DNA was used to exclude SNPs and the discussion raises the important issue of the poor documentation of the reference genome in African countries. If matched GL DNA is not available, have the authors shown absence of these mutations in a panel of normal plasmas? It would only need one or two SNPs to escape filtering to make a big impact on average VAF.

RESPONSE We thank the reviewer for highlighting the important challenge of distinguishing true somatic mutations from single nucleotide polymorphisms (SNPs) in non-coding regions, particularly in the absence of matched germline DNA and given the limited representation of African populations in current reference genomes and SNP databases. Due to cost constraints, we were not able to sequence matched germline samples. To mitigate this limitation, we now implement several stringent filtering strategies in our variant calling (page 7):

1. Panel of normal plasmas: We excluded variants that occurred in ≥ 2 samples in a separate cohort of 12 healthy controls.
2. Population databases: All variants were annotated against gnomAD, and those with a population allele frequency (MAF) $> 1\%$ were excluded.
3. Variant allele frequency (VAF): To minimize the possibility of including germline variants, we excluded all variants with VAF $\geq 40\%$, in addition to the initial exclusion of low-level artefacts with VAF $< 1\%$.
4. Sequencing depth and read support: Only variants with a minimum sequencing depth of 500x and at least 5 mutant reads were retained for downstream analysis.
5. Copy number artefacts: We further excluded 28 variants that showed evidence of being affected by CNAs, as detailed in the previous response.

We acknowledge that despite these steps, a small number of intronic SNPs may still have escaped filtering, particularly in populations underrepresented in global reference databases. We have now added this important caveat to the discussion on page 17, reading as follows:

Distinguishing true somatic mutations from germline single nucleotide polymorphisms (SNPs) in non-coding regions also remains a key challenge in tumour-only sequencing, particularly in African populations that are underrepresented in current reference genomes and SNP databases. Although matched germline DNA was not available in this study, we applied stringent filtering criteria to mitigate this limitation. Nonetheless, we acknowledge that a small number of intronic SNPs may have escaped filtering, highlighting the pressing need for dedicated local genomic reference datasets that are critical in advancing genomic diagnostics and ensuring accurate, population-relevant interpretation of molecular findings in underrepresented African populations.

Following these revisions, we still demonstrate substantial increase in MYC intron 1 mutation counts in BL compared to non-BL cases, although the number of retained variants is substantially lower, with a median count of 21 and 2, in BL and non-BL cases with mutations, respectively (supplementary Fig. 1D). When all BL and non-BL cases were included, including those without detectable mutations, the median MYC intron 1 variant count remained higher in BL (15) than in non-BL (0) (Table 3). We have also expanded our results to show the distribution of MYC intron 1 mutations across different diagnostic categories (benign, HL, DLBCL, and other cancers). Of note, only one benign sample demonstrated mutations in MYC intron 1 (supplementary Fig. 1E).

Given the above difficulty to exclude SNPs in intronic regions, and the frequent missing of IgH rearrangements, how would the model handle a case of acute EBV infection with high EBV titer, splenomegaly and cervical lymphadenopathy? Might this pick up enough factors on the Comprehensive model to score as BL? Since the MYC rearrangement is only found in 48% cases there is no requirement to find this.

RESPONSE: All cases were prospectively enrolled based on the presence of clinical features suggestive of lymphoma consisting of persistent B-Symptoms, lymphadenopathy, jaw swelling or splenomegaly. These show considerable overlap with those of acute EBV infection. Accordingly, our non-BL cases constituted a mix of HL: 36.7% (48/131); benign cases 18.3% (24/131) consisting of: (tuberculous adenitis (5/131) and benign tumours/reactive lymphadenitis including those caused by acute EBV infection (19/131); DLBCL: 16.8% (22/131); and a mixture of other types of cancer 28.2% (37/131). This has now been explicitly stated in the results section, page 11, reading as follows:

Among the 212 participants, where GSP tissue biopsy diagnosis was achieved, 38.2% (81/212) had a diagnosis of BL and 61.8% (131/212) had a non-BL diagnosis. For the non-BL participants, 36.7% (48/131) had HL, 16.8% (22/131) had DLBCL, 18.3% (24/131) had a benign diagnosis (benign tumour, tuberculous adenitis, reactive/EBV-positive lymphadenitis) and the remaining 28.2% (37/131) had a mixture of other types of cancer.

Importantly, we tested for 5 different EBV attributes – the EBV copy numbers per cell (from EBER1, EBER2, EBNA2), EBV max (the highest result of those three for each patient), the proportion of EBV reads out of all sequenced reads (including autosomal reads), the EBV fragment size ratio and the EBV fragment entropy. The comprehensive model integrates both quantitative EBV parameters (EBV DNA copies/mL, EBV max, EBV proportion) and qualitative attributes (fragment size ratio and entropy). Although acute EBV infection is associated with elevated viral copies expressed as EBV DNA copies/mL and EBV max, most BL cases in our cohort demonstrated significantly higher EBV levels (supplementary Fig. 8A).

Besides, all other EBV attributes reflect EBV DNA integrated into the human genome, meaning that they are specific to BL and other EBV-driven malignancies and will not be indicative of acute EBV infection that is characterized by an increase in episomal DNA. In particular, the EBV proportion, the second most important predictor in our model after MYC alterations, is based on the number of EBV reads divided by the number of all reads including also autosomal reads and showed strong differentiation capability between BL and reactive cases. Assessment of the performance of the individual EBV parameters specifically to differentiate BL vs reactive cases identifies EBV proportion (EBVP) and EBV entropy (EBVent) as the best performing attributes (Supplementary Fig. 8A and B).

We acknowledge that assessing model performance in an even larger and even more diverse cohort, including more individuals with acute EBV infection, would further clarify its

specificity in differentiating malignant from reactive EBV-driven processes. To address this concern, the following text has been added to the discussion:
(page 16)

Although acute EBV infection can present with elevated EBV copy numbers, the integration of quantitative and qualitative EBV parameters in our model enables reliable distinction between BL and reactive cases (supplementary Fig. 8).

(page 17)

Although none of the acute EBV infection cases were misclassified as BL, underscoring the model's ability to distinguish malignant from reactive EBV-driven states, the number of acute infection cases was limited. Evaluation in an even larger and more diverse cohort, incorporating additional acute EBV infection samples, would further substantiate the model's specificity and clinical applicability.

Since the MYC rearrangement is detected in only 48% of cases, its absence does not preclude a diagnosis of BL. However, MYC-related features, i.e. MYC translocations and mutations in exon 2 and intron 1 contribute significantly to model performance, as reflected in the importance scores of the comprehensive model. In our initial model, all variables were included irrespective of their interdependencies. During leave-one-out cross-validation, this resulted in coefficient shrinkage for highly correlated variables, making quantitative parameters such as VAF appear statistically more influential than the biologically critical MYC translocation, MYC intron 1 and exon 2 mutations. This behavior is expected in LASSO, a penalized regression framework, which tends to retain only one non-zero coefficient (or a small subset of non-zero coefficients) among a group of highly correlated predictors, regardless of their mechanistic importance. Given that ctDNA and VAF are strongly correlated with MYC mutation status and largely reflect tumor burden rather than specific oncogenic events, we refined the model by excluding these collinear variables in accordance with standard statistical procedures for multicollinearity control. This addition to the methods has been explained under the methods section (page 8) as follows:

Prior to model fitting, candidate predictors were examined for multicollinearity using pairwise correlation and variance inflation factor analysis, and highly collinear variables were excluded to improve model stability and interpretability. Priority was given for those that were most biologically/clinically relevant

There are no other exonic variants in Fig4c and the VAF could be falsely contributed by intronic SNPs. Would the EBV DNA / latency profile in acute EBV infection be sufficiently different from BL to avoid scoring in the comprehensive model? Has the model been tested, and validated as negative, in cases of acute EBV infection without BL?

RESPONSE: We thank the reviewer for this important observation. We included 19 samples from patients with reactive lymphadenitis/ EBV infection in the study to provide biological context and assess model specificity. None of these acute infection cases were misclassified as BL, and their EBV attributes were clearly distinct from those observed in BL (suppl Fig.8A). The comprehensive model was developed using penalized logistic regression (LASSO) and underwent internal leave-one-out cross-validation, which confirmed the model's stability and minimized overfitting. Nevertheless, we recognize that a larger validation cohort will be valuable to further confirm the model's discriminative power in differentiating malignant from acute EBV-associated states. This recommendation has now been added to the Discussion to acknowledge the need for broader validation (page 17), reading as follows:

Obviously, for non-BL cases, standard pathology remains essential for accurately diagnosing other lymphoma subtypes, identifying non-lymphoid malignancies, or

confirming benign lesions. Although none of the acute EBV infection cases were misclassified as BL, underscoring the model's ability to distinguish malignant from reactive EBV-driven states, the number of acute infection cases was limited. Evaluation in an even larger and more diverse cohort, incorporating additional acute EBV infection samples, would further substantiate the model's specificity and clinical applicability.

I am unclear what was included in the gold standard pathology analysis. The authors cite a BJHaem paper, but I can't see what IHC stains were used. It also cites an 81% diagnostic accuracy of that IHC panel. Was the same panel used for GSP in this study? If so, if the gold standard is only 81% accurate, is it meaningful to distinguish between the 5 models tested?

RESPONSE: We have expanded the explanation on the immunohistochemistry (IHC) panel used for the local gold standard pathology analysis in the Materials and Methods section (Local Gold Standard Pathology, page 6), reading as follows:

H&E staining was performed on all samples to evaluate morphology, and IHC was done using the previously described limited IHC antibody panel for BL. This diagnostic approach utilizes a three-phase scoring system. The first phase combines typical Burkitt lymphoma (BL) morphology with immunostains BCL2, CD10, and CD20 to establish a diagnosis. Cases that remain inconclusive proceed to a second phase with CD38, CD44, and Ki67, while unresolved cases undergo a third phase with FISH analysis for MYC-Ig translocation. In our study, we used the first-phase immunostains (BCL2, CD10, and CD20) to support the diagnosis of BL, as this phase has previously been shown to achieve 81% concordance with WHO SoC pathology and is well-suited for the limited-resource settings (Naresh KN et al. British journal of haematology 2011

This limited IHC panel proposed by Naresh et al. (BJHaem, 2011), demonstrates 81% concordance with the comprehensive Standard of Care (SoC) panel that includes BCL2, CD10, CD20, Ki67, CD38, CD44, and FISH for MYC-Ig rearrangement. This limited algorithm represents the accepted SoC in resource-constrained settings and is endorsed by the 5th Edition of the WHO Classification of Haematolymphoid Tumours. Although the diagnostic accuracy of this IHC panel is not absolute, all six models in our study were evaluated against the same consistent gold standard, ensuring that the comparison of model performance remains methodologically sound.

The workings of the models are a black box. The supplementary figure is helpful in understanding the relative contribution of each element of each model, but it's still not completely clear how different factors influence the output. It does seem that the dominant factors to the overall model are EBV proportion. There might be easier, quicker, cheaper ways to measure this – like qPCR. Since this seems to be the main factor contributed by the liquid biopsy, why is a targeted sequencing panel needed at all? Would a similar model using EBV DNA qPCR plus clinical factors be as good? The only two factors that appear to be contributed by the targeted sequencing are the average VAF (see comment 2) and MYC translocation (which is missed in 52% of cases).

RESPONSE: We are grateful for the reviewer's insightful comments, which led to important refinements in our model. Following these changes, the main predictors are now the MYC translocation and the EBV proportion (EBVP), none of which can be measured using standard EBV DNA qPCR. Notably, EBVP was derived as the number of EBV DNA reads divided by the total number of all sequenced reads including autosomal reads (after removal of PCR duplicates), expressed as a proportion. This metric can therefore only be calculated

from sequencing data. The metric that reflects qPCR and has been validated as such (Chamba et al JCO Global Oncology 2025), is the EBV copies per cell (EBER1, EBER2, EBNA2 – or EBVmax (which takes the highest of the three values)). Models that are based on these metrics (EBV, EBV_quantitative) are far less predictive (AUC 0.8 and 0.81; revised Fig. 4) compared to the liquid biopsy model (AUC 0.92).

However, to address the reviewer's suggestion, we also tested the model using EBV DNA copy numbers derived from our sequencing data combined with clinical variables. This model has been incorporated in all analyses and figures (Fig. 4 and Table 1) and we have included a sentence in the result section (page 11) reading as follows

Quantification of EBV in copies per cell, which was previously shown to correlate strongly with EBV DNA qPCR (32), performed inferiorly to the liquid biopsy model both when clinical parameters were included (AUC 0.91 vs. 0.95) and when they were not (AUC 0.80 vs. 0.92).

and in the discussion (page 16),

Combining the liquid biopsy model with clinical parameters yields the highest diagnostic performance with an AUC of 98%, sensitivity of 94% and specificity of 93% for BL detection in the real-world implementation. However, contrary to the model based on quantification of EBV copy/cells, the predictive strength of the comprehensive model is driven by liquid biopsy features, specifically, MYC-IG translocation and EBVP and not by the clinical parameters (supplementary Fig. 7). This underscores the central role of molecular markers in distinguishing BL from other NHLs.

While the EBV_quant_clinical model performed reasonably well thanks to the inclusion of clinical parameters (AUC = 0.91), our comprehensive sequencing model including all liquid biopsy metric achieved higher accuracy and was the only one to demonstrate clinically meaningful (more than 80%) sensitivity and specificity (Se = 86%, Sp = 95%, AUC = 0.95). Importantly, we have also noted in the discussion that although EBV qPCR is useful for distinguishing BL from healthy (EBV positive or negative) controls (Volesky-Avellaneda et al Blood Global Hematology 2025), it is insufficient in clinical practice where multiple EBV-associated pathologies coexist. Our data shows that the inclusion of additional EBV attributes derived from sequencing together with MYC mutation- and translocation-specific markers enhances diagnostic specificity and biological interpretability beyond what EBV quantification alone can provide. This is documented in the discussion (page 16) as follows:

Quantification of plasma EBV DNA has been shown to distinguish BL from healthy individuals, with reported sensitivity of 88% and specificity of 100% (39). In the present study, the EBV quantitative model achieved a lower sensitivity of 57% and a specificity of 95%, likely reflecting the inclusion of other EBV-positive lymphoid malignancies such as HL and DLBCL rather than healthy controls. Similarly, a more recent, larger, multi-country study demonstrated high diagnostic accuracy for EBV DNA quantification using digital droplet PCR but again compared BL cases with healthy population controls, limiting its applicability to real-world clinical settings where multiple EBV-associated malignancies coexist (40). Together, these findings suggest that while EBV quantification is a useful screening tool, it lacks sufficient specificity for diagnostic confirmation. The inclusion of additional EBV attributes in addition to mutation-based and translocation-specific markers therefore enhances diagnostic specificity and biological interpretability beyond what EBV quantification alone can provide and is consistent with previous studies in EBV positive nasopharyngeal carcinoma (26)

The turnaround times in this study are vastly better for liquid biopsy than for IHC. It is not clear to me why this should be the case. In my experience it is surprisingly difficult to achieve this kind of TAT for NGS in a first world context, requiring technical expertise. Why is this easier in Africa? Conversely, why is the IHC TAT so poor? Shown as 6 months in some cases. This is not a criticism of the experimental design - I'm just trying to understand better. Is it that it is easier to train staff to run NGS assays than it is to perform and interpret IHC?

RESPONSE: *The shorter turnaround time (TAT) for the liquid biopsy compared to immunohistochemistry (IHC) reflects differences in workflow, human resource capacity, and infrastructure requirements. As illustrated in Fig. 7, the average one-week TAT for targeted NGS was achieved through an optimized workflow in which cfDNA extraction and library preparation required approximately three days, sequencing one day, and analysis and clinical interpretation another two days. The observed difference in TAT does not mean that NGS is “easier” in Africa, but rather that the liquid biopsy workflow capitalizes on available laboratory personnel, automation, and parallel processes, whereas IHC remains limited by workforce shortages, procedural delays, and dependence on a small number of specialists for interpretation. We have added this important information to the discussion on page 17 - 18, reading as follows:*

It is important to note that the shorter turnaround time (TAT) for the liquid biopsy compared to immunohistochemistry (IHC) reflects differences in workflow, human resource capacity, and infrastructure requirements. Medical laboratory scientists, who are in relatively high supply in Tanzania, already possess foundational skills in molecular biology and sequencing through their work with infectious pathogen surveillance. This existing capacity made it feasible to rapidly train them to perform cfDNA extraction, library preparation, and sequencing steps. Data analysis was streamlined through an automated bioinformatics pipeline, with oversight from a bioinformatician, and results were interpreted by trained clinicians and haematologists using the predefined diagnostic algorithm established in Phase I of the project.

In contrast, the longer TAT observed for tissue-based diagnosis reflects the more constrained human resource pipeline in histopathology. Pathologists are few, and training a specialist takes approximately four years. As previously reported, nearly one-third of diagnostic delay in our setting is procedure-related, primarily due to the time taken to perform and process surgical biopsies (median seven days) (44). While morphological assessment of tissue sections is relatively quick, IHC interpretation is delayed by high pathologist workload and occasional need for repeat biopsies when initial tissue samples are inadequate.

As illustrated in Fig. 7, the average one-week TAT for targeted NGS was achieved through an optimized workflow in which cfDNA extraction and library preparation required approximately three days, sequencing one day, and analysis and clinical interpretation another two days. Medical laboratory scientists, who are in relatively high supply in Tanzania, already possess foundational skills in molecular biology and sequencing through their work with infectious pathogen surveillance. This existing capacity made it feasible to rapidly train them to perform cfDNA extraction, library preparation, and sequencing steps. Data analysis was streamlined through an automated bioinformatics pipeline, with oversight from a bioinformatician, and results were interpreted by trained clinicians and haematologists using the predefined diagnostic algorithm established in Phase I of the project.

Other comments from Editorial Team

1. In addition to addressing the comments of the reviewers, the editorial team feels it is important that the aspects related to implementing the approach within the local healthcare systems are further discussed in the manuscript (discussion section).

RESPONSE: We thank the editor for this important suggestion. We have now expanded the Discussion section to include details of how this liquid biopsy approach could be implemented within local healthcare systems in sub-Saharan Africa. Specifically, we discuss the potential integration of cfDNA testing into existing diagnostic pathways, opportunities for task-shifting and centralised sequencing support, infrastructure and cost considerations, as well as regulatory and capacity-building needs to ensure sustainability and equitable access. These additions are now included in the revised manuscript (Discussion, third from last paragraph, page 18) reading as follows:

Implementing cfDNA-based diagnostics within sub-Saharan African healthcare systems will require alignment with existing diagnostic and clinical pathways. Limited access to histopathology and molecular testing often delays lymphoma diagnosis; a cfDNA assay could complement these workflows by providing a rapid, minimally invasive test deployable from peripheral hospitals and analysed at regional sequencing hubs. Leveraging established molecular networks for HIV or tuberculosis testing offers a practical route for integration while maintaining quality assurance. Implementation research will be key to translate these validated findings into clinical practice, addressing workflow optimization, scalability, and health-system integration of cfDNA testing, as well as its cost-effectiveness and sustainability in routine practice. Pragmatic pilot studies in referral networks could test sample referral systems, centralized sequencing, and digital reporting, while health-economic analyses and stakeholder engagement would inform sustainable financing and regulatory adoption. Together, these efforts would generate the evidence base needed to embed cfDNA diagnostics into national cancer control strategies and improve timely diagnosis and outcomes for lymphoma patients in resource-limited settings.

2. Please also include in the Data Availability Statement links to the repositories where the data are deposited.

RESPONSE: These have been added